# Simultaneous Ground-based and In Situ Swarm Observations of Equatorial F-region Irregularities over Jicamarca

Sharon Aol[1], Stephan Buchert[2], Edward Jurua[1], and Marco Milla[3]

[1]Mbarara University of Science and Technology, Mbarara, Uganda
[2]Swedish Institute of Space Physics, Uppsala, Sweden
[3]Radio Observatorio de Jicamarca, Instituto Geofísico del Perú, Lima, Peru

**Correspondence:** Sharon Aol (saol@must.ac.ug)

**Abstract.** Ionospheric irregularities are a common phenomenon in the low latitude ionosphere. They can be seen in situ as depletions of plasma density, radar plasma plumes, or ionogram spread F by ionosondes. In this paper, we compared simultaneous observations of plasma plumes by the Jicamarca unattended long term investigations of the ionosphere and atmosphere (JULIA) radar, ionogram spread F generated from ionosonde observations installed at the Jicamarca Radio Observatory (JRO), and irregularities observed in situ by Swarm, to determine whether Swarm in situ observations can be used as indicators of the presence of plasma plumes and spread-F on the ground. The study covered the years from 2014 to 2018 when all the data-sets were available. Overall, the results showed that Swarm's in situ density fluctuations on magnetic flux tubes passing over (or near) the JRO may be used as indicators of plasma plumes and spread-F over (or near) the observatory. For Swarm and the ground-based observations, a classification procedure was conducted based on the presence or absence of ionospheric irregularity structures. There was a strong consensus between ground-based observations of irregularity structures and Swarm's depth of disturbance of electron density for most passes. Cases, where irregularity structures were observed on the ground with no apparent variation in the in situ electron density or vice versa, suggest that irregularities may either be localized horizontally or restricted to particular height intervals. The results also showed that the Swarm and ground-based observations of ionospheric irregularities had similar local time statistical trends with the highest occurrence obtained between 20:00 and 22:00 LT. Also, similar seasonal patterns of occurrence of in situ and ground-based ionospheric irregularities were observed with the highest percentage occurrence in December Solstice and Equinoxes and low occurrence in June Solstice. The observed seasonal pattern was explained in terms of the pre-reversal enhancement (PRE) of the vertical plasma drift. Initial findings from this research indicate that fluctuations of in situ density observed meridionally along magnetic field lines passing through JRO can be used as an indication of the existence of well-developed plasma plumes.

**Keywords.** Equatorial Ionosphere, Ionospheric Irregularities

## 1  Introduction

Generally, the ionosphere can be viewed as a layer with a relatively uniform plasma density distribution (Ngwira et al., 2013). However, the nighttime low latitude ionosphere is characterized by localized plasma density structures, known as ionospheric

irregularities (Stolle et al., 2006). The equatorial ionospheric irregularities may be identified as irregular plasma density bite outs observed in situ along Low Earth Orbit (LEO) satellite tracks in the topside ionosphere (Woodman and La Hoz, 1976; Tsunoda, 1980; Tsunoda et al., 1982; Kelley, 2009). The equatorial ionospheric irregularities may also manifest as Equatorial Spread Fs (ESF) which are irregular signatures on ionograms due to backscattering from and above the F-layer at the bottom
(Woodman and La Hoz, 1976; Hysell and Burcham, 1998). Ionospheric irregularities have also been called plasma plumes because of their appearance in range versus time radar displays (Woodman and La Hoz, 1976). The plumes are characterized by elongated, wedge-like cross-sections that extend from the bottom of the F-layer to higher altitudes (Tsunoda, 1980; Tsunoda et al., 1982; Ma and Maruyama, 2006). Equatorial ionospheric irregularities usually extend along magnetic field lines to magnetic latitudes of about $\pm15° - \pm20°$ (Ossakow, 1979; Kil and Heelis, 1998; Nishioka et al., 2008; Kelley, 2009).

Ionospheric irregularities in the low latitudes arise after sunset because of the Rayleigh - Taylor Instability (RTI) which originates from the lower F - region (Woodman and La Hoz, 1976; Kelley, 2009; Schunk and Nagy, 2009). These irregularities vary in scale size, between several centimeters and hundreds of kilometers (Lühr et al., 2014; Xiong et al., 2016; Rino et al., 2016). A width of about 100 km was observed for the depleted ESF bands using an all-sky air-glow imager (Otsuka et al., 2004). The occurrence of ionospheric irregularities varies according to local time, season, longitude, latitude, solar and magnetic activity
(Kil et al., 2009). Their occurrence is a subject of interest because of the effect they have on propagating radio signals. Their presence in the ionosphere may cause amplitude and phase scintillations of radio signals, thereby affecting many applications that rely on these signals (Yeh and Liu, 1982).

Equatorial ionospheric irregularities have been observed many times using ground-based instruments such as incoherent and coherent scatter radars, ionosonde, airglow cameras, and space-based instruments such as rockets and LEO satellites (e.g,
Woodman and La Hoz, 1976; Hysell et al., 1997; Nishioka et al., 2008; Fejer et al., 1999; Burke et al., 2003; Sripathi et al., 2008; Wang et al., 2015; Hickey et al., 2018; Aa et al., 2020). It should be noted that although ionospheric irregularities have been studied extensively, uncertainties still exist in understanding their evolution because of their varying scale sizes (Abdu, 2001; Sripathi et al., 2008; Aa et al., 2020). In this regard, different instruments are limited to observing ionospheric irregularities of particular scale size (Sripathi et al., 2008; Aa et al., 2020). Therefore, coordinated observation of ionospheric
irregularities using different instruments is an effective way to generate an integrated and comprehensive image for specifying ionospheric irregularities of different scale sizes (e.g, Sripathi et al., 2008; Cherniak et al., 2019; Aa et al., 2019, 2020). Particularly, the Jicamarca Radio Observatory (JRO) has provided a rare opportunity to observe ionospheric irregularities using multiple ground-based instruments because of its strategic location (12.0°S, 76.8°W; magnetic latitude 0.6°S) at the magnetic equator. Many studies have reported the connection between scintillation producing irregularity structures observed
by JULIA and Equatorial Plasma Bubbles (EPBs) observed in situ over Jicamarca (e.g, Morse et al., 1977; Basu et al., 1980; Hysell and Burcham, 2002; Burke et al., 2003; Kelley et al., 2009; Siefring et al., 2009; Roddy et al., 2010; Nishioka et al., 2011, etc). However, the relationship between meter-scale irregularities detected by coherent scatter radars to the underlying state parameters of the ionospheric plasma is not yet well understood (Hysell et al., 2009). Previous studies (e.g, Kelley et al., 2009; Siefring et al., 2009; Hysell et al., 2009; Roddy et al., 2010; Nishioka et al., 2011, etc) have mostly compared zonally
oriented in situ plasma density measurements from Communication Navigation Outage Forecasting System (C/NOFS) satellite

with JULIA observations. The European Space Agency's (ESA's) Swarm satellites revisit neatly the JRO in orbits oriented in the meridional direction, providing a renewed opportunity to study in situ ionospheric irregularities recorded by Swarm in the meridional direction in comparison with observations from Jicamarca.

A quantitative statistical relationship between plasma bubbles observed in situ in the meridional direction, 250 MHz amplitude scintillation, and JULIA observations were reported by Burke et al. (2003) using data recorded by the polar-orbiting Defense Meteorological Satellite Program (DMSP). From the observations made by Burke et al. (2004), the plasma plumes recorded by JULIA frequently occurred at altitudes lower than that of the DMSP orbit. Most of the plasma plumes failed to reach altitudes >600 km and therefore, they were not observed by DMSP satellite which orbited at an altitude of about 840 km. It is possible to compare in situ measurements made by Swarm and JULIA observation at altitudes of 460 km (Swarm A and C) and 510 km (Swarm B). Compared to DMSP, Swarm allows a comparison of measurements from identical instruments at different altitudes and in different longitudinal sectors (Zakharenkova et al., 2016). Previous comparison of Swarm in situ measurements with ground-based radar observations (e.g, Zakharenkova et al., 2016) mostly used LP measurements at 2 Hz frequency. The faceplate carried by Swarm as part of the Electric Field Instrument (EFI) has enabled the discovery of small-scale (down to 500 m length scale along the space-craft track) ionospheric irregularities. Also, the previous comparison of Swarm in situ measurements with ground-based radar observations was mostly a single case presentation. Zakharenkova et al. (2016) demonstrated that the gradual spatial separation between Swarm A, C and B would decrease the likelihood that all three satellites could capture ionospheric irregularity signatures in the same localized region or near a particular equipment installed on the ground such as ionosonde, radars, etc. Nevertheless, only one single case of comparison between Swarm and JULIA observation was provided by Zakharenkova et al. (2016).

In this paper, we quantitatively compared the in situ observations of ionospheric irregularities recorded by the Swarm satellites with ground-based measurements of plasma plumes made by the JULIA radar for the years from 2014 to 2018, to determine whether Swarm in situ observations can be used as indicators of the presence of plasma plumes and spread-F on the ground. The comparison is complemented with ionosonde measurements of spread-F over JRO. Booker and Wells (1938) observed echoes on ionograms from ionosonde observations and proposed that these echo signatures were originating from ionosphere disturbances. As far as we know, Wang et al. (2015) were among the first to make concurrent observations of strong range spread-F and ionospheric irregularities measured in situ using ROCSAT-1 satellite and they found that strong spread-F were caused by the ionospheric irregularities. However, ROCSAT-1 orbited at about 600 km altitude with 35° orbital inclination. Therefore, we also compared the JULIA and Swarm observations of ionospheric irregularities with spread-F signatures recorded by an ionosonde colocated with the JULIA radar. To understand the range of altitude above sea level where ionospheric irregularities occur and the effect they have on ground observations, a comparison of in situ electron density variation with ground-based measurements over a long time is essential.

This paper is organized in the following order: In Sect. 2, the data and methods used in this study are described. In Sect. 3, the results are presented and discussed. The findings of this study are summarized in Sect. 4.

## 2 Data and Methods

### 2.1 Data

This section provides brief descriptions of the instruments and data-sets used in this study to examine the signatures of ionospheric irregularities. In this study, we analyzed data obtained from Swarm, JULIA radar and ionosonde.

### 2.1.1 Swarm Measurements of Electron Density

The Swarm mission consists of three polar-orbiting satellites i.e., Swarm A, B, and C (Friis-Christensen et al., 2006). They were launched in near-polar orbits at an initial altitude of about 500 km on 22 November 2013 (Xiong et al., 2016; Wan et al., 2018). Each satellite is equipped with an Electric Field Instrument (EFI) which is mounted on the ram side of the spacecraft (Friis-Christensen et al., 2006; Knudsen et al., 2017). The ion density is derived from the EFI faceplate current assuming that the current is carried by ions hitting the faceplate due to the orbital motion of the spacecraft (Buchert, 2016). However, due to quasi-neutrality $N_i$ must be equal to the electron density $N_e$. With the 16 Hz $N_e$ measurements, Swarm observes irregularity structures with scale sizes of up to 500 m. Friis-Christensen et al. (2006) and Knudsen et al. (2017) provide detailed information on Swarm and onboard instruments. In this study we used, the 16 Hz $N_e$ measurements to examine topside irregularity structures. The faceplate $N_e$ data is readily available at `http://earth.esa.int/swarm`.

### 2.1.2 The JULIA Radar

The JULIA radar is a PC-based system for data acquisition. JULIA uses low-power transmitters with a frequency of approximately 50 MHz and Jicamarca's main antenna (Hysell and Burcham, 1998, 2002; Burke et al., 2003). Its aim is to record ionospheric irregularities and neutral atmospheric waves at the equatorial region for long periods of time. The pulse width used in the JULIA experiments during the period of study was 25 $\mu$s and the pulse repetition was 160 pulses per second. In addition, 248 range gates of 3.75 km separation were sampled, going from 0 km to about 930 km during the period of study. For the identification of 3 m-scale ionospheric irregularities, the back-scattered 50 MHz JULIA radar echo was used. The radar observations provide the Signal to Noise Ratio (SNR), Doppler velocity and spectral width as a function of height and time. Data collected by the JULIA radar is readily available at `http://jro.igp.gob.pe/madrigal/`. The website has JULIA data from 1996 till to-date. Our analysis was restricted to the years from 2014 to 2018 when JULIA, Swarm, and ionosonde data-sets were available.

### 2.1.3 The Digital Ionosonde

The Equatorial Spread F (ESF) signatures are often recorded by ionosondes installed in the JRO. The ionosonde at the JRO is a Digisonde DPS-4 (Reinisch et al., 1998) which records ionograms (altitude versus frequency plots) at 15 min intervals. The Automatic Real-Time Ionogram Scaler with True height (ARTIST) UML ionogram autoscaling tool is used to scale the ionograms and the outputs are plasma frequency profiles versus altitude (Reinisch et al., 2005; Zhang et al., 2015). The

equatorial spread-F signatures were analyzed from the ionograms. The ionosonde data are also available on the Madrigal website and we used all the Jicamarca ionograms for the years from 2014-2018.

## 2.2  Methods

This section presents the analysis techniques used in this study to identify the observed ionospheric irregularity signatures
using Swarm, JULIA and ionosonde.

### 2.2.1  In situ Ionospheric Irregularity Identification

To examine topside ionospheric irregularities, the 16 Hz $N_e$ Swarm faceplate data were used. We followed the same method as Ngwira et al. (2013), Huang et al. (2014), and Aol et al. (2020) to derive the absolute electron density perturbation along Swarm orbital tracks, but we focused mainly on small-scale equatorial plasma structures. We utilized a 2-s running mean filter
to determine the mean $N_e$. The selected running mean is equivalent to a 15 km scale length, given Swarm's velocity of about 7.5 km/s. The mean $N_e$ was subtracted from the original observations to get the residual, $\Delta N_e$ , similar to Ngwira et al. (2013) who obtained the residual using Total Electron Content (TEC) data. The standard deviation of the residuals was then calculated at a running window of 2-s to represent the magnitude of the perturbation (std($\Delta N_e$)). There is no standard threshold definition of how large std($\Delta N_e$) must be to identify plasma irregularities (Huang et al., 2014; Wan et al., 2018). However, the period
considered in this study was characterized by low solar activity and the recorded ionospheric irregularities were very weak. Therefore, a threshold value of std($\Delta N_e$)=$1 \times 10^{10}$ m$^{-3}$, similar to that adopted by Huang et al. (2014), has been selected to provide a reasonable irregularity event identification at small scales and the relatively low Swarm altitudes during the study period.

### 2.2.2  Ground-based Ionospheric Irregularity Identification

The JULIA system computes and stores measurements of the zeroth and first lags of the auto-correlation function (ACF) of the signals from two receivers connected to the east and west quarters of the main antenna (Smith et al., 2015; Zhan et al., 2018). The total power, Doppler velocity at first moment, and Doppler spectral width of the scattering signals can be determined from these measurements (Hysell et al., 1997; Hysell and Burcham, 1998). Of particular interest was the SNR measurements derived by the JULIA system to check plasma plumes for a given evening. Besides, we also used the vertical plasma drift
measurements made by the Jicamarca Incoherent Scatter Radar (ISR) to examine the pre-reversal enhancement (PRE) drifts (Fejer et al., 1996; Smith et al., 2016). Field-aligned irregularities in the F-region are often observed between 1800 LT and 0600 LT by the JULIA radar  (Hysell and Burcham, 1998; Smith et al., 2016). Therefore, to compare the Swarm observations with the JULIA measurements, only swarm satellite passes for the time between  1800 LT and 0600 LT were considered.

The comparison of JULIA and Swarm observations were supplemented with ionosonde measurements from the JRO. The
ionosonde data analysis was carried out using the SAO Explorer software (Reinisch et al., 2005). To display ionograms, both the raw and processed (SAO) data were loaded into the SAO Explorer software. In addition, the spread-F index QF, known as

the mean spread of the diffusing F layer trace, was obtained by the ARTIST directly from the ionograms using the SAO explorer (Galkin et al., 2008; Zhang et al., 2015) and this was also used in this study to analyze the spread-F signatures. The spread-F index QF is defined as the extent of the diffuse reflection in km averaged over all frequencies where a diffuse echo appeared. For simplicity, the virtual height is used at each frequency to determine the range extent of the reflection. The ARTIST software for data analysis is described by Galkin et al. (2008). Spread F ionograms were similarly studied by Abdu et al. (2012) using magnetically conjugate ionosondes in South America, and by Zhang et al. (2015) for ionosondes and scintillation receivers at Sanya. In the following section, the results of this study are presented and discussed.

## 3 Results and Discussions

### 3.1 Observations of Ionospheric Irregularities

Examples of equatorial ionospheric irregularity events observed by Swarm A, C on 2015-03-09 and B on 2015-04-05 are shown in Fig. 1. From the left panel of Fig. 1, Swarm A and C encountered ionospheric irregularity structures along their

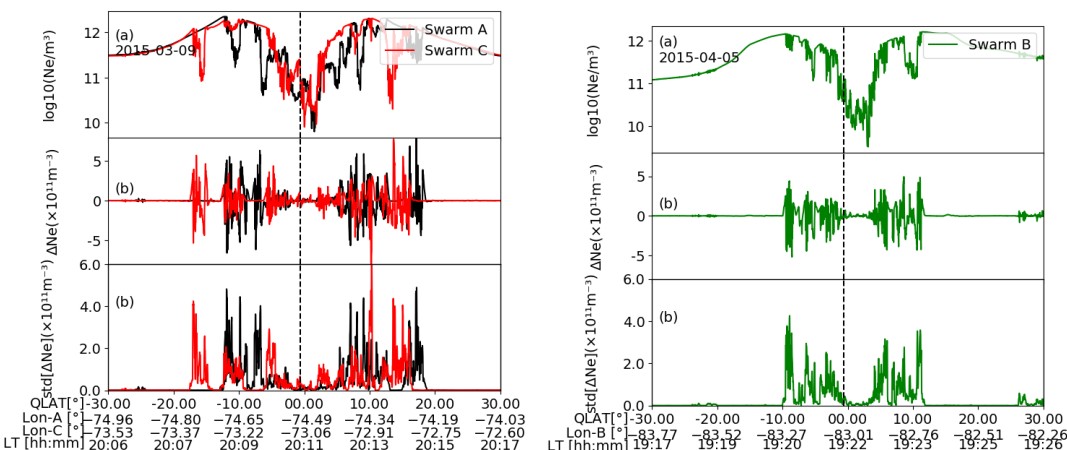

**Figure 1.** Swarm Faceplate $N_e$ data of ionospheric irregularity events on 2015-03-09 and 2015-04-05. The panels show: (a) the $N_e$ variation, (b) $\Delta N_e$, and (c) std($\Delta N_e$) as functions of QLat, longitude (LON), and local time (LT). The dashed vertical black line represents the approximate QLat of JRO.

tracks occurring between about $\pm 10° - \pm 20°$ quasi-dipole latitude (QLat) (Laundal and Richmond, 2016), while they orbited over JRO on 2015-03-09. Also, Swarm B which orbited at about 510 km altitude above sea level recorded ionospheric irregularity structures on 2015-04-05 as seen from the right panel of Fig. 1. Zakharenkova et al. (2016) also observed large $N_e$ depletions along Swarm passes using the 2 Hz $N_e$ measurements made by the Langmuir Probe (LP) in comparison with JULIA radar observations. With the 16 Hz data, $N_e$ depletions can also be observed at even smaller scales down to 500 m (Aol et al., 2020). Panel (b) of Fig. 1 presents the $\Delta N_e$ with background variations subtracted. Figure 1(c) shows how well the quantified

absolute density perturbation captured the small-scale ionospheric irregularities in the faceplate $N_e$ measurements. Given the fact that the irregularity structures were observed in situ at Swarm altitudes, this shows that these irregularity structures were in the topside ionosphere. The observed ionospheric structures occurred post-sunset and they are most likely because of the generalized RTI (Kelley, 2009).

The coherent scatter radar observations of ionospheric plasma irregularities are often shown in Range-Time-Intensity format in which the SNR is plotted against altitude (range) and time (Woodman and La Hoz, 1976; Hysell and Burcham, 1998). The major categories of plasma plumes that have been observed by the JULIA radar are Bottom-type, Bottom-side, and Topside (e.g, Woodman and La Hoz, 1976; Hysell and Burcham, 1998). Examples of these categories are presented in Fig. 2, which shows Bottom-type, Bottom-side, and Topside structures in panels (a) to (c), respectively. In general, from Fig. 2, the observed

structures are visible mostly post-sunset and this coincides with the time when the generalized RTI is expected to intensify (Kelley, 2009). From panel (a) of Fig 2, Bottom-type structures are weak and narrow scattering layers and their thickness is

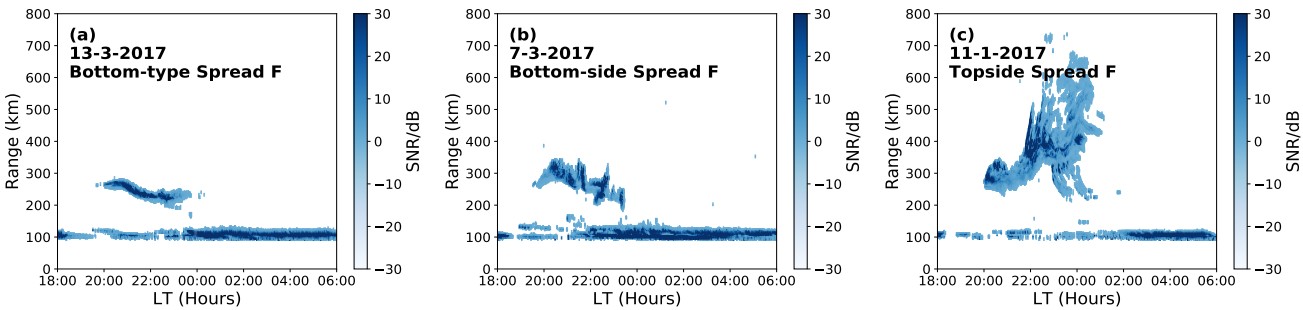

**Figure 2.** Examples of the different types of ESFs that may be observed by the JULIA radar. The color bar presents the signal to noise ratio (SNR) in dB.

less than about 50 km. Bottom-type structures are too weak to induce prominent ionogram spread F or cause intense radio scintillation at VHF frequencies and above (Hysell, 2000). Their disturbance in the ionosphere is also not sufficient to cause signatures on air-glows (Hysell, 2000). Bottom-side structures correspond to broad, more structured, and stronger scattering

layers at relatively higher altitudes that last for a few hours as seen in the middle panel of Fig 2, while Topside layers or radar plumes (see panel (c) of Fig. 2) represent larger-scale elongated structures originating from bottom-side layers and extending to the topside ionosphere (Hysell and Burcham, 1998; Hysell, 2000; Chapagain et al., 2009; Chapagain, 2011). They are indicators of strong plasma plumes (Smith et al., 2016).

To check the altitude coverage of the various types of plumes observed by the JULIA radar compared to the Swarm altitudes,

a histogram of percentage occurrence of maximum heights was generated for the different types of plumes. To determine the plume maximum height, SNR outliers were first eliminated to minimize spurious data points. The maximum height then

corresponds to the maximum range in km where SNR was recorded. Figure 3 shows the frequency of occurrence of the maximum height achieved by the various types of plumes for the years from 2014 to 2018. From Fig. 3, the Swarm altitude

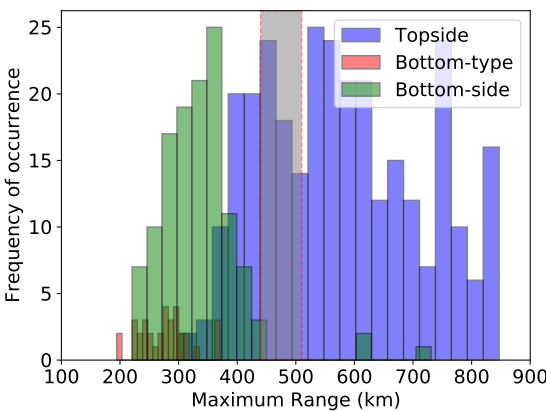

**Figure 3.** Frequency of occurrence of the maximum height achieved by the different types of ESFs observed by JULIA radar for the years from 2014 to 2018. The gray region indicates the approximate altitude coverage of the Swarm satellites from 2014 to 2018.

range coincides with high frequency of occurrence of maximum range of topside plasma plumes. This reveals that the Swarm orbits are most suitable to detect topside plasma plumes compared to the other types. The following subsection presents in situ observations of ionospheric irregularities by Swarm over or near the JRO longitude in comparison with the JULIA and ionosonde observations.

## 3.2 Coincident Ground-based and Swarm Observation of Ionospheric Irregularities

Here, in comparison with the ground-based observations, selected Swarm orbits that were directly overhead or passed close to the JRO are presented with observed plasma density structures. Figure 4 shows example cases on 2015-03-02 and 2015-03-08 where Swarm A and C passed directly over and near JRO, respectively. Column (i) of Fig. 4 shows the Range-Time-Intensity maps laid over with Swarm A and C positions. The JULIA radar started to detect weak 3-m irregularities from an altitude of about 300 km at about 1930 LT on 2015-03-02 and 2015-03-08. Gradually, the irregularities evolved into a series of spectacular plume structures that extended to altitudes of about 800 km. The plumes were only visible in the pre-midnight hours and this corresponds to the time when the RTI dominates (Kelley, 2009; Schunk and Nagy, 2009). The observed plasma plumes coincided with the Swarm passes.

Columns (ii) and (iii) of Fig. 4 show that Swarm encountered irregularity structures along their tracks on 2015-03-02. The irregularities were more intense near the Equatorial Ionization Anomaly (EIA) belts at about $\pm 10° - \pm 15°$ QLat than at the quasi-dipole equator. Zakharenkova et al. (2016) analyzed one sample of plasma plumes recorded by JULIA and from the results they presented, several Global Positioning System (GPS) satellites orbiting over the JRO encountered irregularities near the magnetic equator on 2015-03-02. Therefore, the structures observed by Swarm could be associated with the JULIA plasma

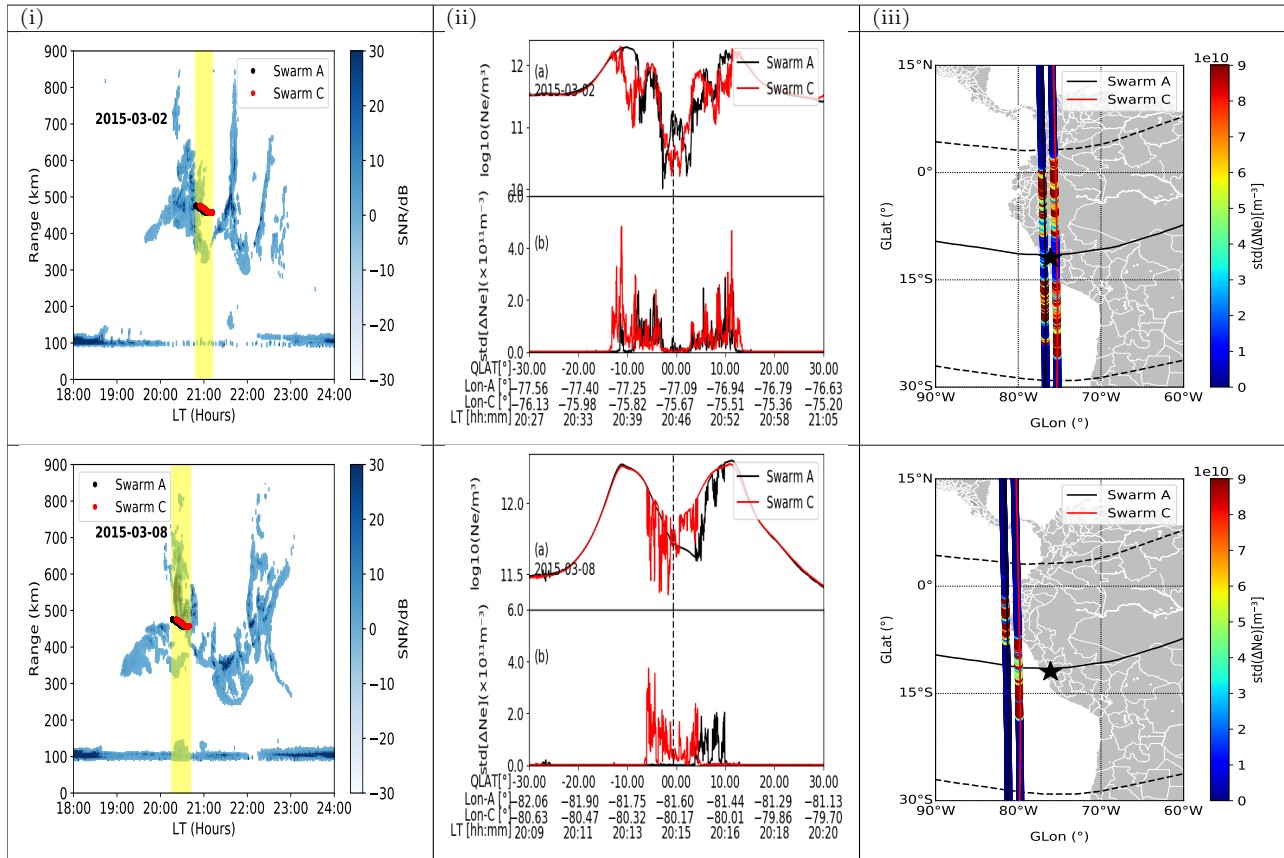

**Figure 4.** Examples of collocated observations by Swarm and JULIA radar on 2015-03-02 and 2015-03-08. The local time coverage and the corresponding altitude of Swarm while orbiting over or near JRO are (black for Swarm A, red for Swarm C) shaded yellow in column (i). The QLat of JRO is indicated with a vertical dotted black line in column (ii). The ground tracks of Swarm and the location of JRO are shown in the maps in column (iii). The thick black line in column (iii) shows the geomagnetic equator, while the dotted black lines show the EIA belts ($\pm 15°$ QLat).

plumes on 2015-03-02. Roddy et al. (2010) and Nishioka et al. (2011) also presented single case events while comparing in situ plasma density measurements made by C/NOFS satellite with JULIA observations. However, in the results presented by Roddy et al. (2010) and Nishioka et al. (2011), the EIA could not be resolved because C/NOFS orbited in a nearly meridional direction.

5  On 2015-03-08, Swarm A and C crossed the quasi-dipole equator in the evening sector at geographic longitudes of about 81.6°W and 80.17°W, respectively. The $N_e$ profiles of Swarm A and C in columns (ii) and (iii) of Fig. 4 show depletions near the quasi-dipole equator. However, the longitudes of the Swarm satellites were offset from the JRO longitude to the west.

Burke et al. (2003) made a similar observation, comparing DMSP plasma density measurements with JULIA observations. Ionospheric irregularities are generally assumed to drift westward across the magnetic field lines (Kelley, 2009). Therefore, the depletions met by Swarm A and C may not correspond to the plumes observed by the JULIA radar. The irregularity structures observed by Swarm on 2015-03-08 may correspond to the plume remnants that drifted across the radar beam.

We also checked on the Spread F signatures on ionosonde data from JRO in comparison with the results presented in Fig. 4. Figures 5 and 6 show ionograms produced on 2015-03-02 and 2015-03-08 using the SAO explorer. The ionosonde measurements were recorded at 15 min intervals. The sequence of ionograms presented in Fig. 5 show that spread Fs were

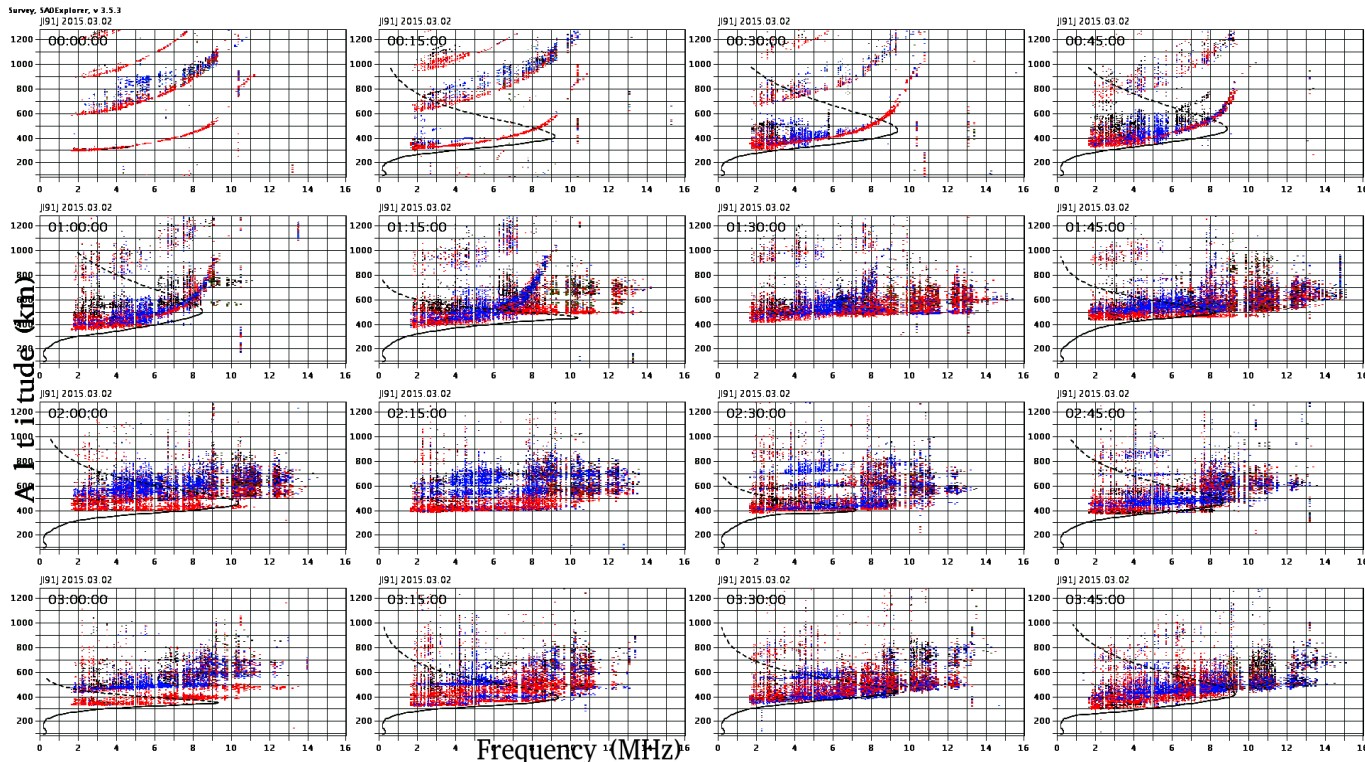

**Figure 5.** Ionograms showing occurrence of ESF on 2015-03-02 from 00:15 UT to 03:45 UT (LT= UT-5hrs).

continuously observed from 00:30 UT-05:00 UT (19:30 LT-24:00 LT), while Swarm encountered ionospheric irregularities on 2015-03-02 between about  20:27 LT to 21:05 LT (see Fig. 4). The ionograms on 2015-03-08 also showed strong spread Fs
starting at 00:15 UT(19:15 LT) and this coincided with the time period when ionospheric irregularities and plasma plumes were recorded by Swarm and JULIA, respectively. The results presented in Fig. 4, 5, and 6 show that the in situ ionospheric irregularities, spread Fs, and plumes were observed over and near JRO simultaneously. Strong range spread F is caused by ionospheric irregularities and can, therefore, be regarded as a result of the generalized RTI mechanism (Rastogi et al., 1989; Wang et al., 2008; Shi et al., 2011; Alfonsi et al., 2013). The Spread F signatures are triggered by irregularities at the bottom or
within a growing plasma bubble or by declining bubbles (Abdu et al., 2012). Figures 4, 5, and 6 provide evidence that JULIA,

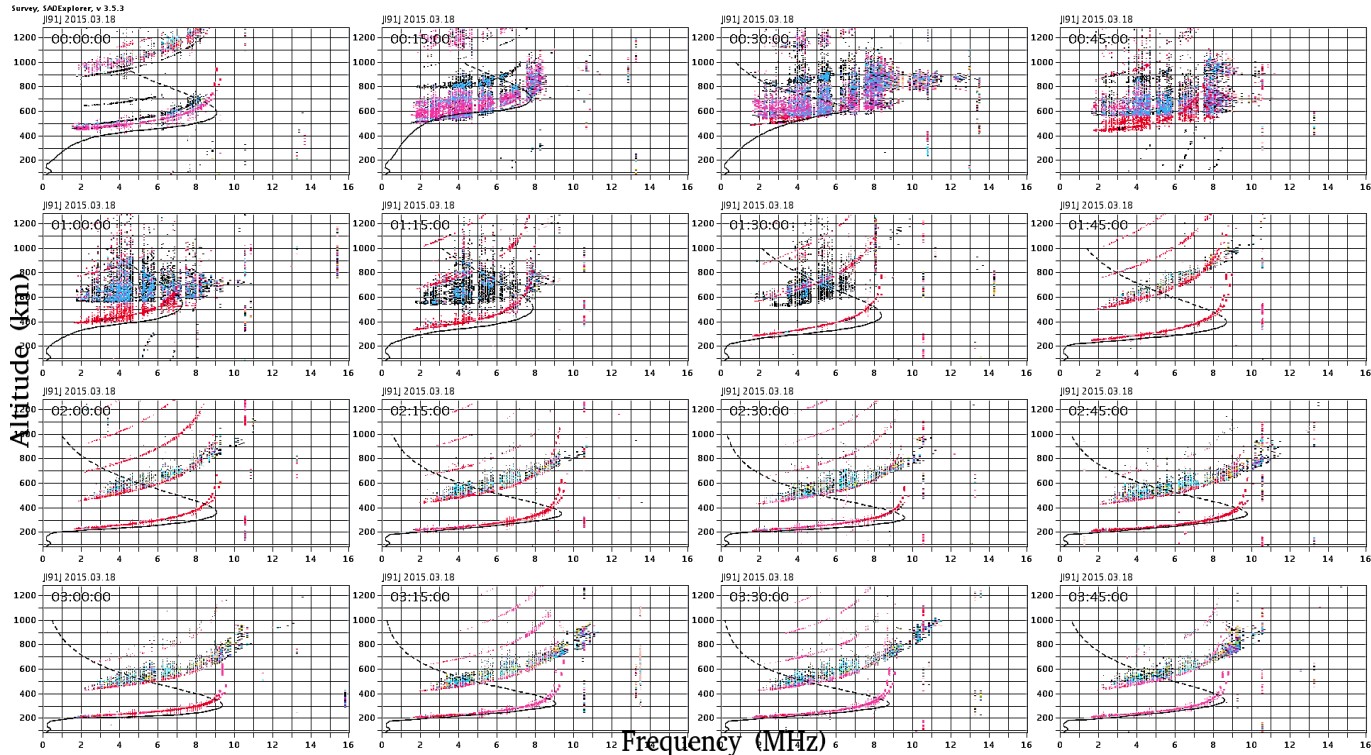

**Figure 6.** Ionograms showing occurrence of ESF on 2015-03-08 from 00:15 UT to 03:45 UT (LT= UT-5hrs).

Swarm, and ionosonde simultaneously observed ionospheric irregularities over the JRO. In the next section, we present the results of a statistical analysis of Swarm, JULIA, and ionosonde observations.

### 3.3 Statistical Analysis of Occurrence of Ionospheric Irregularities

The formation of equatorial ionospheric irregularities is influenced by several factors including local time, season, magnetic
5 latitude and longitude. The data sets accumulated for the years from 2014 to 2018 were sufficient to compare the dependence of ground-based and in situ occurrence of ionospheric irregularities on various factors. Here, we present the results of statistical analysis carried out in this study. The specific details of each statistical result are described in the following subsections.

#### 3.3.1 Statistics of Occurrence of Ionospheric Irregularities by Category

The Swarm satellites regress in longitude by about 22.5° between orbital ascending nodes. Therefore, in comparison with
10 JULIA and ionosonde data, the Swarm passes were allowed to be within ±5° magnetic longitude of the JRO to make sure that a sufficient amount of Swarm passes could be used for the statistical examination. Both JULIA and ionosonde data during the time when Swarm was within ±5° longitudinal range from the longitude of the ground site were selected. Summary plots such as those presented in Fig. 4 were generated for all days during the years from 2014 to 2018 for which the data was available. In

total, 560 night-time orbits were used for which JULIA, Swarm, and ionosonde data were available concurrently. The outputs of the summary plots could be categorized into four cases considering the presence (or not) of irregularities. In general, these four cases are: Irregularities observed both on the ground and in situ, No irregularities observed both on the ground and in situ, Irregularities observed only in situ, and Irregularities observed only on the ground. For each Range-Time-Intensity plot, the SNR corresponding to the peak height was determined and an event was identified as a significant irregularity when the peak height was $\geq 400$ km. For peak height less than 400 km, these were classified as weak or no irregularities. It is important to note that the peak height less than 400 km is a representation of bottom-type, bottom-side, and no equatorial spread Fs and therefore, all spread-F altitudes were taken into consideration during the analysis. For the in situ Swarm observations, we considered a threshold of $1 \times 10^{10}$ m$^{-3}$ for std($\Delta N_e$) as a significant irregularity event, while for std($\Delta N_e$) less than the threshold were considered as no irregularities. For the ionosonde measurements, QF values greater than or equal to 20 km were considered as significant irregularity events. For each category, the percentage occurrence was computed as a ratio of the total number events in that category to the number of observations. These cases are presented in Fig. 7 for (a) Swarm and JULIA and (b) Swarm and ionosonde.

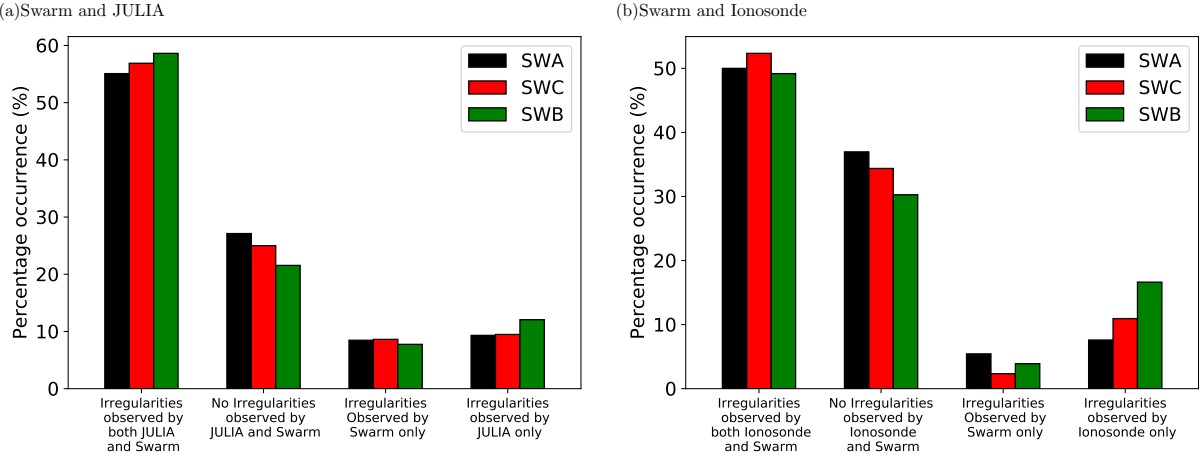

**Figure 7.** Percentage occurrence of irregularities in each category observed by the (a) Swarm satellites and JULIA observations and (b) Swarm satellites and ionosonde observations for the years from $2014 - 2018$.

In about 55.08% of the cases for Swarm A, 56.89% for Swarm C, and 58.62% of the cases for Swarm B, irregularity structures were detected by Swarm and JULIA as seen from panel (a) of Fig. 7. In about 27.12% of the cases for Swarm A, 25.0% for Swarm C, and 21.55% of the cases for Swarm B, no irregularity structures were detected by the Swarm satellites and JULIA. In panel (b) of Fig. 7, a high percentage occurrence was also observed when there was agreement (irregularities observed by both ionosonde and Swarm & no irregularities observed by ionosonde and Swarm) between Swarm and ionosonde. The

two categories where there was an agreement in panels (a) and (b) indicate that Swarm satellites, JULIA, and ionosonde simultaneously observed ionospheric irregularities. Burke et al. (2003) also examined the relationship between measurements of JULIA and DMSP satellite during 110 nights for the years from 1998 to 1999. In comparison to the statistical results presented in Fig. 7 for Swarm and JULIA, the DMSP satellite sampled very few EPBs than the JULIA radar detected plumes. According to Burke et al. (2003), there was a low probability that JULIA and DMSP would encounter ionospheric irregularity structures because most plumes could not ascend to altitudes greater than 600 km. There were also some disagreements between the ground-based and space observations where JULIA and ionosonde detected plume structures, while Swarm registered no events as seen from the statistical results in panels (a) and (b) of Fig. 7. For these cases, the Swarm altitudes during the pass were examined. It was observed that the plume structures did not ascended to Swarm altitudes by the time the satellites passed over Jicamarca or the satellites were simply in a different location. For instances when Swarm registered events, while JULIA and ionosonde recorded no signatures, we checked on the longitudinal separation between the satellite passes and the ground-site. The longitudinal separations obtained between the Swarm passes and the ground site were often $\approx 5°$ and the magnitude of the in situ perturbations were relatively low. Ionospheric irregularities tend to be magnetic field aligned (Ossakow, 1979; Kil and Heelis, 1998; Nishioka et al., 2008; Kelley, 2009) and therefore, Swarm may encounter irregularities in situ of relatively small magnitudes, while JULIA and ionosonde do not identify any events, for wider longitudinal offset of a pass from the ground site.

Zakharenkova et al. (2016) showed two cases of Swarm passes over JULIA, i.e., one case when Swarm A encountered ionospheric irregularities and JULIA recorded Spread F, and another case when Swarm B never registered any irregularity structure and JULIA never recorded any spread F. The statistical results shown in Fig. 7 assert that Swarm B can also detect irregularities and plasma bubbles associated with plumes and spread F, but with more mismatches than for Swarm A/C. Swarm B recorded more mismatches than A/C because of the progressive temporal and altitudinal separation between Swarm B and A/C (Zakharenkova et al., 2016). Swarm B orbits at a higher altitude compared to A/C and it crosses the same region later than A/C. Generally, in Fig. 7, a difference in percentage occurrence in all categories is observed between Swarm A and C although they orbit at the same altitude above sea-level. The large scale longitudinal bubble structure is sometimes observed with the two Swarm satellites (Xiong et al., 2016), but for small scale irregularities, the 1.5° longitudinal separation between the satellites is too large for a significant correlation between them.

### 3.3.2 Local time and Seasonal Variation of Ionospheric Irregularities

Numerous studies have shown that irregularity structures at low latitudes are a post-sunset phenomenon owing to the electrodynamics launched after sunset (e.g, Stolle et al., 2006; Lühr et al., 2014; Abdu et al., 2012). Here, we also compared the Local time dependence of occurrence of plasma plumes observed by the JULIA radar, spread-F recorded by the ionosonde, and small-scale ionospheric irregularities encountered by Swarm for different seasons. Figure 8 shows the percentage occurrence of plasma plumes as a function of local time and height grouped into different seasons i.e December solstice, June solstice, March equinox, and September Equinox. To obtain the results presented in Fig. 8, ground-based JULIA SNR data for the years from 2014 to 2018 were used. To eliminate the impact of geomagnetically disturbed conditions on the statistical outcomes, the data

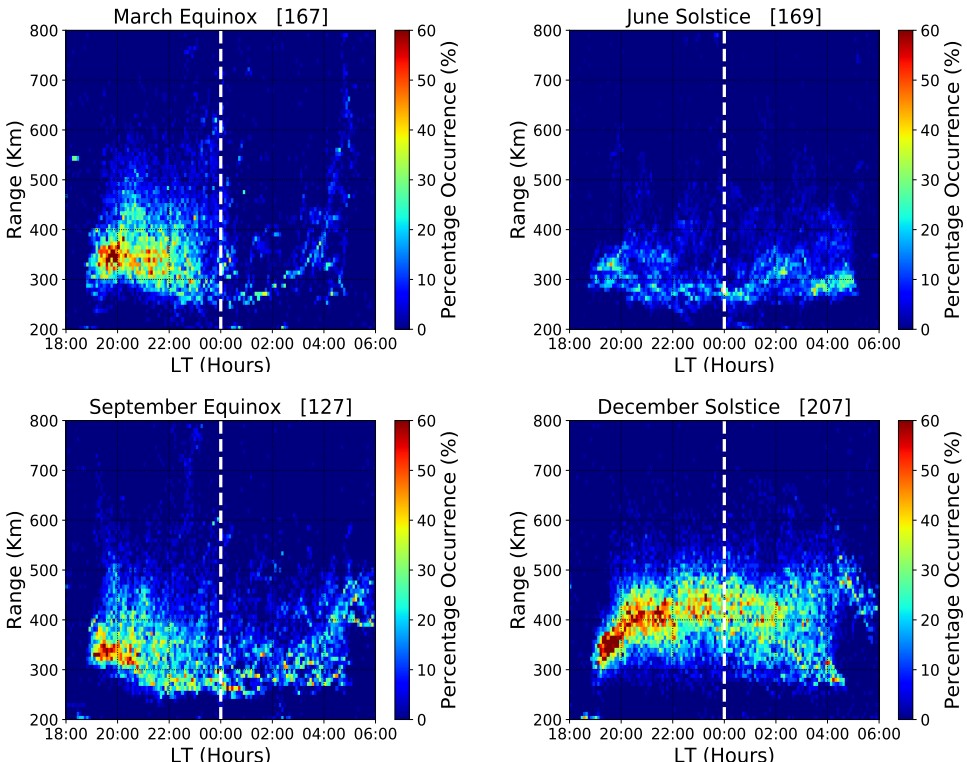

**Figure 8.** Percentage occurrence plasma plumes as a function of local time and height for the years $2014 - 2018$. Each panel represents a season. The number indicated in brackets is a count of days used to generate each season's statistics when measurements were made. The dotted white vertical line represents midnight.

were filtered and only those recorded during quiet geomagnetic conditions ($\text{Kp} \leq 3$) were taken into account. The JULIA data accumulated for the years from 2014 to 2018 were sufficient for examining the seasonal variation. Therefore, the seasonal dependence of local time distribution of JULIA observations of ionospheric irregularities was also examined by grouping all the data into different seasons corresponding to March Equinox (Feb-Mar-Apr), June Solstice (May-Jun-Jul), September Equinox

5  (Aug-Sep-Oct), and December Solstice (Nov-Dec-Jan). For each local time-height bin, the percentage occurrence was obtained by dividing the number of observations with $\text{SNR} > 10 \text{ dB}$ by the total number of observations (e.g, Smith et al., 2016).

It is visible from Fig. 8 that the plasma plumes only occurred at night. Fig. 8 shows the occurrence of irregularities in plasma plumes starting at about 1900 LT and generally lasts past midnight. This observation is similar to those of previous studies (e.g, Kil and Heelis, 1998; Hysell and Burcham, 2002; Smith et al., 2016). The observed plasma plumes are connected with

10  the nonlinear development of the RTI, which is initiated at the bottom of the F region. (Woodman and La Hoz, 1976; Huang, 2018). In December Solstice and the Equinoxes, the highest percentage occurrence occurs. The lowest percentage occurrence is observed in June Solstice. The daily variations of the vertical plasma drift measured by the ISR were used to better understand

the seasonal patterns observed in Fig. 8. Figure 9 presents the Local time variation of F region vertical drift velocity for the years from 2014 to 2018. From Fig. 9, the PRE of the vertical plasma drifts can be seen around sunset hours (between 1700

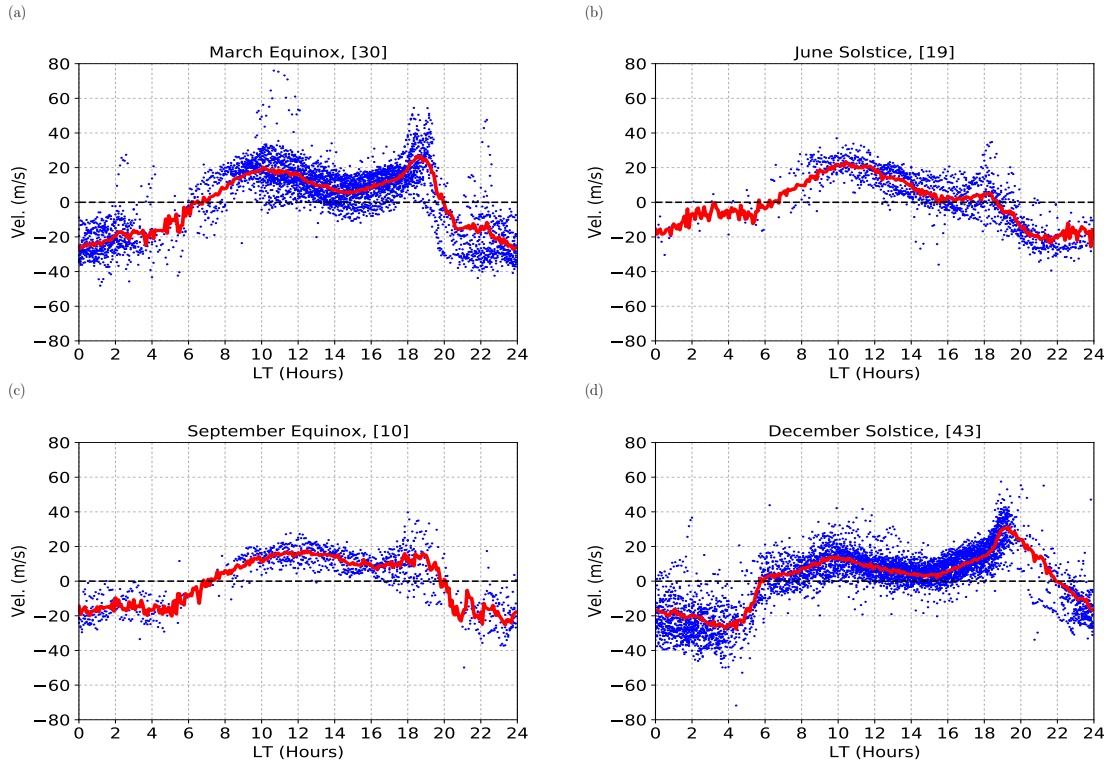

**Figure 9.** Local time variation of F- region vertical plasma drifts as a function of local time. The red curves represent the averaged vertical drift curves. Each panel represents a season.

LT and 2000 LT) before its reversal. Figure 9 shows the highest PRE peak in December Solstice and Equinox seasons, while the PRE peak is the lowest on June Solstice. Similar observations were made by Smith et al. (2016). Comparing the results

5    presented in Fig. 9 with the local time distributions presented in Fig. 8, it follows that high occurrence of post-sunset topside spread F is associated with enhancements of the PRE peak. A high PRE implies significant $E \times B$ vertical drifts which boost the rate of RTI growth (Sultan, 1996; Fejer et al., 1999). The PRE moves the ionospheric F layer to higher altitudes where there is less interaction between ions and neutrals.The decreased interaction between ions and neutrals leads to increased RTI (Jayachandran et al., 1993; Kelley, 2009).

10    Figure 8 also shows a relatively high occurrence of irregularities after midnight especially in the solstices and September Equinox. Similar observations were made by Hysell and Burcham (2002) and Smith et al. (2016). The extension of the occurrence of plumes post-midnight may be due to the late reversal time and small post-reversal electric fields (Hysell and Burcham,

2002). Despite the use of transmitters of low power, the JULIA radar can also detect weak post-midnight irregularities, particularly during the solstice seasons as shown in Fig. 8. However, the detected post-midnight ionospheric irregularities often exist at much lower altitudes than the ones presented by Smith et al. (2016). Using Jicamarca radar measurements, Fejer et al. (1999) reported that the ionospheric irregularities that occur after midnight are typically well-formed structures that can be connected

to the disturbed dynamo.

Figure 10 shows the QF indices derived from ionosonde observations as a function of local time and months. To obtain the results presented in Fig. 10, ground-based ionosonde data for the years from 2014 to 2018 were also used. The data were also filtered and only those recorded during quiet geomagnetic conditions ($Kp \leq 3$) were considered. Considering that equatorial

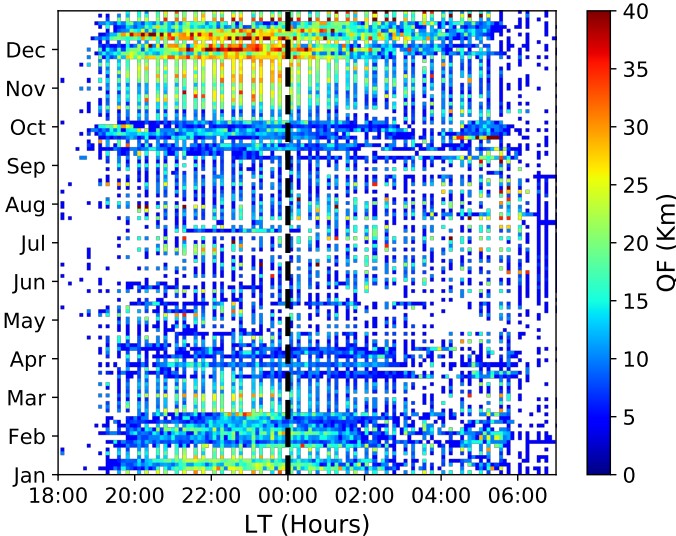

**Figure 10.** Month and local time (LT) variations of QF indices observed over JRO for the years from 2014-2018. The white spaces show periods when no spread F was detected. The dotted black vertical line represents midnight.

ionospheric irregularities are nighttime phenomena, we only presented QF data from 18:00 LT to 06:00 LT in Fig. 10. To

generate Figure 10, for each month (y-axis), the QF were averaged over 0.1 hr Local time bins. Figure 10 shows that the QF index was high in the post-sunset period with peak values occurring between about 20:00 LT and 00:00 LT in December solstice and the equinoxes. The high QF values in the equinoxes and December solstice is most likely due to the RTI, which is usually triggered at the bottom side of a rising equatorial F layer. The rate of growth of the RTI depends on the meridional wind and the eastward electric field PRE determined by the longitudinal gradient of flux-tube-integrated conductivity (Sultan, 1996;

Basu, 2002). In June solstice, QF values were small in the after sunset as seen in Fig. 10. Generally, Fig. 10 shows moderate QF values lasting till local midnight or longer (similar to the trend presented in Fig. 8). The low post-sunset QF values in June Solstice can be attributed to the small PRE and this can also be seen in panel (c) of Fig. 9.

(a) Swarm A

(b) Swarm C

(c) Swarm B

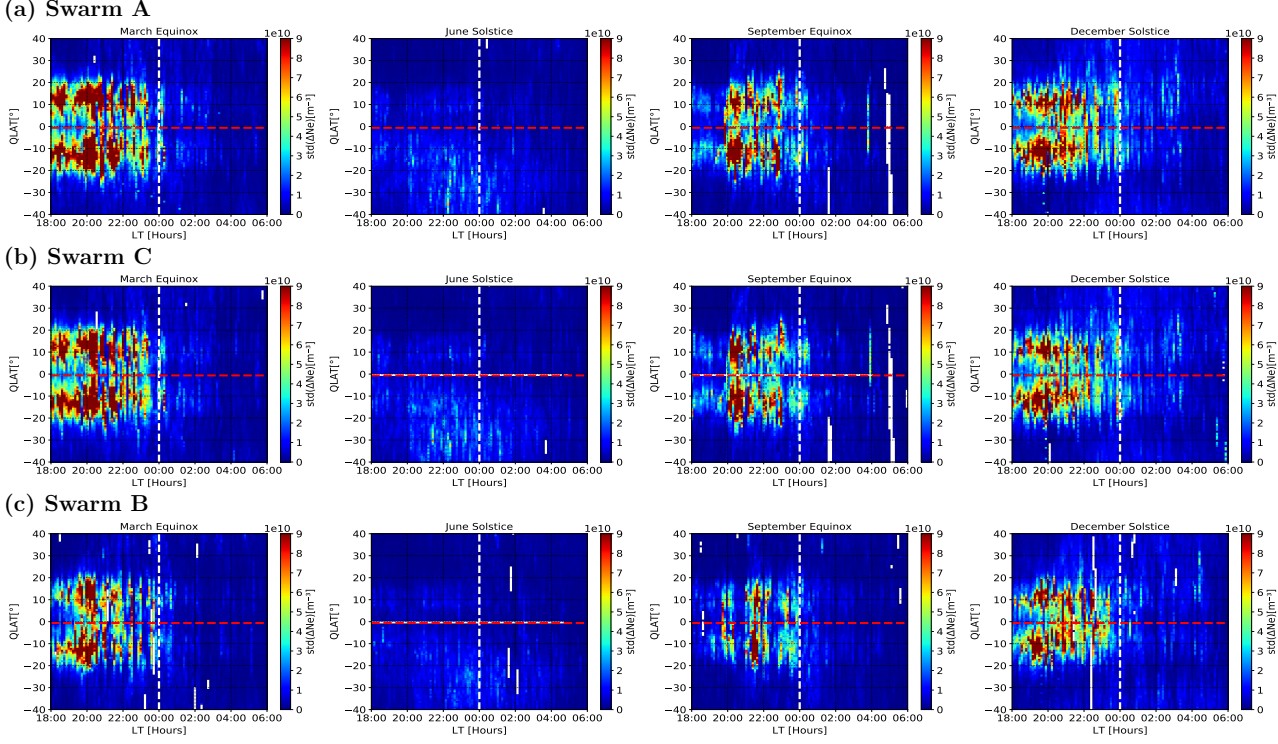

**Figure 11.** Quasi-dipole latitude and local time distributions of std($\Delta N_e$) for Swarm A, C and B for the years from $2014 - 2018$. The dotted white vertical line represents midnight, while the dotted red horizontal line represents the quasi-dipole latitude of Jicamarca ($\approx -0.6°$). The white spaces represent data gaps.

Figure 11 shows the QLat-LT distributions of std($\Delta N_e$) for the Swarm satellites for the years from 2014 to 2018. Recall that the Swarm passes were allowed to be within $\pm5°$ magnetic longitude. The $N_e$ data collected for the five years were also grouped into different seasons similar to those presented in Fig. 8. The results presented in Fig. 11 were also generated considering only the geomagnetically quiet conditions (Kp $\leq$ 3). The std($\Delta N_e$) was then calculated in bins of $1° \times 0.1$ hr resolution in QLat and Local time. The occurrence rate of ionospheric irregularities does not always correspond to the highest amplitude of irregularity structures (Wan et al., 2018). Therefore, we presented the calculated std($\Delta N_e$) per bin as a function of QLat and Local time as seen in Fig. 11. From Fig. 11, high std($\Delta N_e$) values frequently occurred between about $\pm10° - \pm20°$ QLat i.e., at the approximate location of the EIA belts. The distribution of the std($\Delta N_e$) as seen in Fig. 11 is essentially symmetrical about the quasi-dipole equator. The symmetrical distribution about the magnetic equator has also been observed in earlier studies (e.g, Stolle et al., 2006; Carter et al., 2013; Wan et al., 2018, etc) and this confirms that equatorial ionospheric irregularities usually extend along the magnetic field lines in the north and south directions and they are concentrated at the

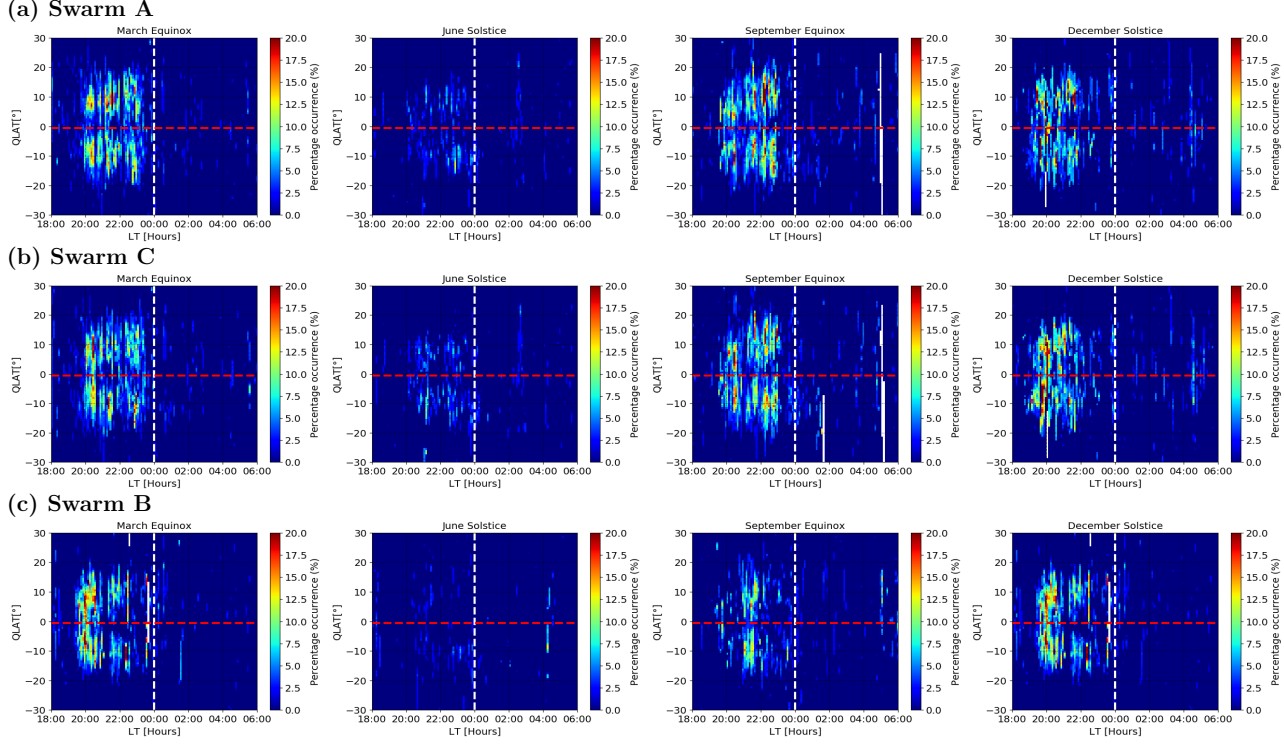

**Figure 12.** Quasi-dipole latitude and local time distributions of IBI for Swarm A, C and B for the years from $2014 - 2018$. The dotted white vertical line represents midnight, while the dotted red horizontal line represents the quasi-dipole latitude of Jicamarca ($\approx -0.6°$). The white spaces represent data gaps.

EIA belts (Kelley, 2009). The std($\Delta N_e$) attained a maximum between 20:00 LT and 22:00 LT. A decrease was detected after 22:00 LT until 06:00 LT. The local time distribution of std($\Delta N_e$) for Swarm A, B and C is the same as that of the quiet-time F-region echoes presented in Fig. 8 and the QF distribution presented in Fig. 10. The distribution of std($\Delta N_e$) shown in Fig. 11 has peak values at local times and QLat ranges where the RTI is expected. In terms of seasons, as observed from Fig. 11,

5   high values of std($\Delta N_e$) were seen in the equinoxes and December solstice, while the lowest values were detected in June solstice. This is similar to the seasonal dependence of quiet-time F-region echoes presented in Fig. 8. From Fig. 11, Swarm hardly encountered post-midnight irregularities while orbiting over South America during all the seasons. From Fig. 8, it is observed that the post-midnight plumes often existed at lower altitudes in the solstices and September Equinox. Therefore, the low post-midnight ionospheric irregularity observations by Swarm may be because the plumes failed to reach Swarm altitudes.

10   For comparison, Fig. 12 shows the quasi-dipole versus local time distribution of ionospheric irregularities based on the Ionospheric Bubble Index (IBI), which is a standard Level 2 product of the Swarm mission (Park et al., 2013). The IBI provides

information on climatology of ionospheric irregularities and the level of magnetic field disturbance by taking both the electron density and magnetic field measurements into account (Park et al., 2013; Wan et al., 2018). It is important to note that the IBI is just either 1, 0, -1 for bubble detected, not detected or undetermined, respectively. Therefore, to generate the results in Fig. 12, the data-sets were first grouped into different seasons corresponding to March Equinox, June Solstice, September Equinox, and December Solstice. For each season, the data was then binned into $1° \times 0.1$ hr quasi-dipole latitude-local time bin. For each quasi-dipole latitude-local time bin, the percentage occurrence was obtained by dividing the number of observations with $IBI = 1$ by the total number of observations. The binned percentage occurrence of $IBI = 1$ shown in Fig. 12 has similar seasonal characteristics as the in situ irregularities shown in Fig. 11. The percentage occurrence of $IBI = 1$ ranges from 0 to 20% which is relatively low. The low percentage occurrence of ionospheric irregularities derived from IBI was also observed by Wan et al. (2018). This may be because the magnitude of ionospheric irregularities must be large enough to cause magnetic field fluctuations (Wan et al., 2018). The latitudinal profile of std($\Delta N_e$) and percentage occurrence of $IBI = 1$ have peaks near the anomaly crests (about $\pm 15°$ QLat). The diamagnetic effect in fluctuations of $N_e$ are believed to be the cause of $IBI = 1$ and this occurs at the anomaly crests (Stolle et al., 2006; Lühr et al., 2003).

The Bragg condition for the JULIA radar implies that the coherent Spread echoes are from density variations at about 3 m wavelength (Kelley, 2009). The Bragg condition for backscatter means that a radar can only observe structures in the refractive index with size close to the half radar wavelength (Kelley, 2009; Hocking et al., 2016). The Swarm electron density measurements used in this study are limited by the sampling rate of 16 Hz and the orbital velocity of about 7.5 km/s to wavelengths of about 500 m, respectively, and longer. The good correlation between the Swarm measurements at these wavelengths and the Spread echoes from Swarm altitudes suggest that the irregularities seen by Swarm occur over a spectrum of different wavelengths, at least from about 500 m down to the radar wavelength of 3 m. A non-linear decay of unstable waves could explain this. In addition, we expect that the radar signal could be affected by scintillations which are particularly known from one-way signal propagation such as in GNSS and VHF satellite beacons (Burke et al., 2003; Zuo et al., 2016). For the radar, this could be relevant for echoes where the Bragg reflection occurs at high altitudes, above Swarm. Fresnel theory shows that wavelengths at the Fresnel scale of $\sqrt{(2\lambda d)}$ are most relevant for causing scintillations. $\lambda$ is the wavelength, 3 m for the JULIA radar, and $d$ the distance from the perturbation to the receiver, 445-510 km for Swarm. This gives a Fresnel scale between about 1.6 km and 1.7 km. The Swarm measurements generally indicate that irregularities at such scales are present near the paths of Bragg reflected radar signals. We, therefore, suggest that Spread F signals may at times be a result of both the Bragg backscattering at the highest altitude as well as scintillations of the radio waves to and from the scatter region.

## 4 Conclusions

In this paper, the results of a study of equatorial ionospheric irregularities detected by the JULIA radar and ionosonde in comparison with in situ $N_e$ measurements made by Swarm for the years from 2014 to 2018 was presented. Cases of coincidence between Swarm, JULIA, and ionosonde observations were discussed. Also, the JULIA, ionosonde, and Swarm observations

were examined statistically during geomagnetically quiet conditions. The local time and seasonal statistical patterns obtained from JULIA, ionosonde. and Swarm were explained using drift measurements by the ISR.

Results based on the JULIA radar and ionosonde agreed with the plasma density obtained from measurements of the Swarm faceplate for single satellite passes over or near the JRO. Basing on an on-off classification, in the majority of cases, when the JULIA radar detected topside plasma plumes, Swarm also observed plasma bubbles when its trajectory crossed directly overhead or near the JRO. This was also true for the ionosonde measurements. A few exceptions were also observed when the JULIA radar and ionosonde detected the presence of plasma structures, while Swarm did not record any bubbles and vice versa. For the case when JULIA and ionosonde recorded irregularity signatures, while Swarm observed no structures, the plume structures may not have ascended to Swarm altitudes by the time the satellites passed over Jicamarca or the satellites were simply in a different location. Swarm was able to detect ionospheric irregularities in situ, while no signature was recorded on the ground simply because the irregularities occurred at magnetic longitudes which were largely offset from the longitude of the ground-site. Statistical differences between Swarm A and C were observed and these were attributed to the 1.5° longitudinal separation between them which becomes significant for small scale irregularities.

The three phenomena i.e., plasma plumes observed by the JULIA radar, spread-Fs recorded by the ionosonde, and small scale irregularities detected by Swarm revealed similarities in the patterns of occurrence basing on local time and different seasons. The highest occurrence rate was observed in December Solstice and in the Equinoxes, while a low occurrence rate was observed in June Solstice. Measurements of the vertical plasma drift, made by the ISR, were used to understand the seasonal dependence of the occurrence of topside spread F and in situ density irregularities. The seasonal dependence of the occurrence of topside spread F and in situ $N_e$ irregularities can be explained by the extent to which plasma is drifted vertically upwards.

The five years of Swarm faceplate $N_e$ data set has revealed a lot of detailed features of electron density variations associated with plasma bubbles. In situ measurements of $N_e$ by Swarm are a promising tool to indicate a likelihood of plasma plumes and spread F occurrence at times and locations where radar/ionosonde is not available. The geometry, however, is an important factor and therefore, in determining whether satellite observations are valid or accurate for any given ground site, algorithms should take into account the position of the satellite and the apex height of the magnetic field lines.

*Data availability.* The official website of Swarm is `http://earth.esa.int/swarm` and `ftp://swarm-diss.eo.esa.int` is the server for the distribution of Swarm data. The IBI measurements used in this study can be obtained from `https://swarm-diss.eo.esa.int/#swarm%2FLevel2daily%2FLatest_baselines%2FIBI`. The radar measurements used in this study can be obtained from the Madrigal database at `http://jro.igp.gob.pe/madrigal/`. The Kp index used in this study were obtained from the website `http://omniweb.gsfc.nasa.gov/`. The SAO Explorer software was obtained from `http://ulcar.uml.edu/SAO-X/SAO-X.html`.

*Author contributions.* The ideas presented in this manuscript were designed and implemented by Aol Sharon, Stephan Buchert, Edward Jurua, and Marco Milla. With contributions from all the co-authors, Aol Sharon prepared the manuscript.

*Competing interests.* The authors declare that there is no competing interest.

*Acknowledgements.* This study was financially supported by the International Science Program (ISP) of the Uppsala University, Sweden and by the Jicamarca International Research Experience Program (JIREP, `http://www.igp.gob.pe/convocatorias/jirep`) of the Instituto Geofisico del Peru with support from the US NSF. The authors acknowledge the ESA Swarm team for the Swarm mission. The Jicamarca Radio Observatory is a facility of the Instituto Geofisico del Peru operated with support from the NSF AGS-1732209 through Cornell University. We also acknowledge the anonymous referees for their insightful comments which significantly improved the quality of this paper.

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
