# Peer review of "Simultaneous Ground-based and In Situ Swarm Observations of Equatorial F-region Irregularities over Jicamarca"

_Annales Geophysicae, 2019_

## Referee Comment (RC1) · Anonymous Referee #1 · 8 Jan 2020

Review notes:

Present work describes comparison of ionospheric irregularitiy data between the satellite SWARM electron densities and groundbased JULIA radar data and ionosonde data, using the data obtained over Peru from 2014 to 2018. In terms of identification of ionospheric irregularities, they found good similarities between the three different data set. Their final conclusion was that SWARM data can be used as a tool to indicate presence of plasma plumes and Spread F occurrence. The data sets used in the manuscript are interesting and comparison of them in terms of plasma bubble finding, or validation study for identification of the plasma irregularities, are useful. However, if one asks

what is a new finding in this work I could not find anything. The authors could have a new aspect if they further discuss in the data analysis, I guess. Therefore my conclusion is that the present work will not be acceptable as a scientific paper without any further scientific new finding.

Minor comments are as follows: Page 4, line 29, "Ngwira et al. (2013a)": change to "(Ngwira et al., 2013a) Page 5, line 8, "Smith et al„": change to "(Smith et al., 2015; Zhan et al. 2018) Page 5, line 29, "from the left panel of Fig. 1": change to "from the right panel of Fig. 1" Page 7, Figure 3: Please explain why the authors plotted the maximum ranges as a function of time (days). It seems to have no relation between the maximum range and Time(days). Page 12, Figure 7: The authors did not discuss in the case of "Irregularities observed by SWARM only". According to the height coverage of Julia radar (90 to 800 km), any irregularities detected by SWARM should be detected by Julia radar. It could be difference of threshold of detection amplitude (?), further discussion would be helpful for readers. Page 16, line 1, "The Bragg condition": Please explain what is "Bragg condition" for readers who are not familiar to the phrase. Page 17, Conclusions, lines 13-14: "A few exceptions were also observed when„": Further scientific discussion would be valuable for readers.

---

## Author Comment (AC1) · 31 Jan 2020

**Simultaneous Ground-based and In Situ Swarm Observations of Equatorial F-region Irregularities over Jicamarca**

As authors of the manuscript angeo-2019-153, we thank the anonymous referee for the constructive suggestions and comments. In enhancing the quality of the paper, all the remarks we received on this research will be taken into consideration and we present our response to each of them individually below. For the convenience of the referee we have repeated in the response the relevant comments and then given texts we intend to add in the revised manuscript in blue.

**Response to Anonymous Referee #1 comments**

**Comment:** Present work describes comparison of ionospheric irregularitiy data between the satellite SWARM electron densities and groundbased JULIA radar data and ionosonde data,using the data obtained over Peru from 2014 to 2018. In terms of identification of ionospheric irregularities, they found good similarities between the three different data set. Their final conclusion was that SWARM data can be used as a tool to indicate presence of plasma plumes and Spread F occurrence. The data sets used in the manuscript are interesting and comparison of them in terms of plasma bubble finding, or validation study for identification of the plasma irregularities, are useful. However, if one asks what is a new finding in this work I could not find anything. The authors could have a new aspect if they further discuss in the data analysis, I guess. Therefore my conclusion is that the present work will not be acceptable as a scientific paper without any further scientific new finding.

**Response:** We are glad that the referee finds the data sets used in our paper interesting and comparison of them in terms of plasma bubble finding, or validation study for identification of the plasma irregularities useful.

The main focus, findings and innovations may not have been demonstrated and highlighted clearly in our manuscript. In the revised version, we are going to improve our article organization and put more description to highlight our findings. Below are the points we want to address in this study:

– The main focus of this study is to determine whether Swarm in situ observations can be used as indicators of the presence of plasma plumes and spread-F on the ground by comparing simultaneous observations of plasma plumes by the Jicamarca unattended long term investigations of the ionosphere and atmosphere (JULIA) radar, ionogram spread F generated from ionosonde observations installed at the Jicamarca Radio Observatory (JRO), and irregularities observed in situ by Swarm. The combined multi-instrument measurements provide a more integrated and comprehensive way to study the morphological structure, development, and seeding mechanism of ionospheric irregularities.

– Most previous studies (e.g, Kelley et al., 2009; Siefring et al., 2009; Hysell et al., 2009; Roddy et al., 2010; Nishioka et al., 2011) have mostly compared zonally oriented in situ plasma density measurements from Communication Navigation Outage Forecasting System (C/NOFS) satellite with JULIA observations. The Swarm satellites revisit neatly the same area in orbits oriented in the merdional direction. Therefore, our study compares sub-kilometer in situ ionospheric irregularities recorded by Swarm in the meridional direction with observations from Jicamarca. We found that the results based on the JULIA radar and ionosonde agreed with the plasma density obtained from measurements of the Swarm faceplate for single satellite passes over or near the JRO.

– Previous comparison of Swarm in situ measurements with ground-based radar observations (Zakharenkova et al., 2016, e.g,) mostly used LP measurements at 2 Hz frequency. The faceplate carried by Swarm as part of the Electric Field Instrument (EFI) has enabled the discovery of small-scale (down to 500 km length scale along the spacecraft track) ionospheric irregularities. In this study, we used Swarm faceplate measurements at a frequency of 16

Hz. Coherent scatter radars e.g, the JULIA radar can monitor irregularities at high spatial resolution (3 m scale length for the case of JULIA) and therefore these were comared with Sarm faceplate observations of ionospheric irregularities of small scales. The high-resolution faceplate data enabled smaller scale structures to be identified in electron density. Also, previous comparison of Swarm in situ measurements with ground-based radar observations (Zakharenkova et al., 2016, e.g,) were mostly single case presentations. Our study provides an extended statistical analysis covering years from 2014 to 2018.

– As far as we know, a quantitative statistical relationship between plasma bubbles observed in situ in the meridional direction, 250 MHz amplitude scintillation, and JULIA observations were reported by Burke et al. (2003) using data recorded by the polar-orbiting Defense Meteorological Satellite Program (DMSP). However, DMSP orbited at an altitude of about 840 km and this was a limitation in that most ionospheric irregularities didnot ascend to DMSP altitude. In our study, we used the polar orbiting Swarm satellites which has provided a renewed opportunity to compare in situ and JULIA observation at altitudes of 460 km (Swarm A and C) and 510 km (Swarm B). Compared to DMSP, Swarm allows comparison of measurements from identical instruments at different altitudes and in different longitudinal sectors.

– As far as we know, Wang et al. (2014) were the first to make concurrent observations of strong range spread-F and ionospheric irregularities measured in situ using ROCSAT-1 satellite and they found that strong spread-F were caused by the ionospheric irregularities. However, ROCSAT-1 orbited at about 600 km altitude with 35° orbital inclination. Therefore, we also compared the JULIA and Swarm observations of ionospheric irregularities with spread-F signatures recorded by an ionosonde colocated with the JULIA radar.

We are going to highlight these points as listed above in the revised manuscript.

**Minor Comments:**

**Comment1:** Page 4, line 29, "Ngwira et al. (2013a)": change to"(Ngwira et al., 2013a)

**Response:** The citation identified by the referee will be changed as suggested.

**Comment2:** Page 5, line 8, "Smith et al,,,,": change to "(Smith et al., 2015;Zhan et al. 2018)

**Response:** The citations identified by the referee will be changed as suggested.

**Comment3:** Page 5, line 29, "from the left panel of Fig. 1": change to "fromthe right panel of Fig. 1"

**Response:** The sentence will be changed as suggested by the referee.

**Comment4:** Page 7, Figure 3: Please explain why the authors plotted themaximum ranges as a function of time (days). It seems to have no relation between themaximum range and Time(days).

**Response:** The maximum ranges were plotted to check the altitude coverage of the various types of plumes observed by the JULIA radar compared to the Swarm altitudes. The maximum ranges were obtained for each "day of the month" when the JULIA data was available. As identified by the referee, there indeed seems not be a direct relation between "Maximum range" and "Time (days)". Instead each maximum range corresponds to "a day of the month". Therefore, the xlabel of Figure 3 will be changed to "Day of the month".

**Comment5:** Page 12, Figure 7: The authors did not discuss in the case of "Irregularities observed by SWARM only". According to the height coverage of Julia radar (90 to 800 km), any irregularities detected by SWARM should be detectedby Julia radar. It could be difference of threshold of detection amplitude (?), further discussion would be helpful for readers.

**Response:** Ideally, any irregularity detected by Swarm would be detected by JULIA depending on the proximity of the Swarm pass to the JULIA longitude and the magnitude of the irregularitiy. Given the longitudinal range (±5°) that was used for good statistical coverage (see Pg 10 Line 3), there were chances that Swarm would record ionospheric irregularities without JULIA identifying any plume structures. This is because ionospheric irregularities tend to be magnetic field aligned (Ossakow, 1979; Kil and Heelis, 1998; Nishioka et al., 2008; Kelley, 2009).

[Figure]

**Figure 1.** Example of collocated observation by Swarm and JULIA on 2015-11-08.

An example is presented here in Figure 1. On the day 2015-11-08, Swarm encountered ionospheric irregularities, while JULIA recorded no plume structures. Clearly, the Swarm passes were offset from the JULIA longitude to the east. The following discussion will be added in the revised manuscript, to explain "Irregularities observed by SWARM only"

For instances when Swarm registered events, while JULIA and ionosonde recorded no signatures, we checked on the longitudinal separation between the satellite passes and the ground-site. The longitudinal separations obtained between the Swarm passes and the ground site were often $\approx 5°$ and the magnitude of the in situ perturbations were relatively low. Ionospheric irregularities tend to be magnetic field aligned (Ossakow, 1979; Kil and Heelis, 1998; Nishioka et al., 2008; Kelley, 2009) and therefore, Swarm may encounter irregularities in situ of relatively low magnitudes, while JULIA and ionosonde do not identify any events, for wider longitudinal offset of a pass from the ground site.

**Comment6:** Page 16, line 1, "The Bragg condition": Please explain what is "Bragg condition" for readers who are not familiar to the phrase.

**Response:** Bragg-Scattering explains the effect of periodic or quasi-periodic variations of the refractive index on the propagation of electromagnetic waves when the scales of the variations are of the order of the wavelength. The radio waves fulfilling the Bragg condition scatter at periodic variations of the refractive index such that a coherent superposition of reflections occurs and a reflected waves propagates in the direction of the radar receiver. The Bragg condition describes when the constructive interference occurs in a certain ratio between the transmitted wavelength and the distance of the reflective sub-surfaces:

$d = \frac{\lambda_t}{2.\cos\theta}$

$d$=distance between maxima and minima of the refractive index
$\lambda_t$=transmitted wavelength
$\theta$=incident angle

The following sentences will be included in the manuscript as further explanation of the "Bragg condition":

The Bragg condition for backscatter means that a radar can only observe structures in the refractive index with scale sizes close to the half radar wavelength (Kelley, 2009; Hocking et al., 2016).

**Comment7:** Page17, Conclusions, lines 13-14: "A few exceptions were also observed when,,,": Furtherscientific discussion would be valuable for readers

**Response:** This has been addressed in response to **Comment5**.

In addition, the following sentences will also be included in the conclusion as further discussion for the observation made in lines 13-14:

For the case when JULIA and ionosonde recorded irregularity signatures, while Swarm observed no structures, the plume structures may not have ascended to Swarm altitudes by the time the satellites passed over Jicamarca or the satellites were simply in a different location. Swarm was able to detect ionospheric irregularities in situ, while no signature was recorded on the ground because the irregularities occurred at magnetic longitudes which were largely offset from the longitude of the ground-site.

**References**

Burke, W. J., Huang, C. Y., Valladares, C. E., Machuzak, J. S., Gentile, L. C., and Sultan, P. J.: Multipoint observations of equatorial plasma bubbles, Journal of Geophysical Research (Space Physics), 108, 1221, https://doi.org/10.1029/2002JA009382, 2003.

Hocking, W. K., Röttger, J., Palmer, R. D., Sato, T., and Chilson, P. B.: Atmospheric radar: Application and science of MST radars in the Earth's mesosphere, stratosphere, troposphere, and weakly ionized regions, Cambridge University Press, 2016.

Hysell, D. L., Hedden, R. B., Chau, J. L., Galindo, F. R., Roddy, P. A., and Pfaff, R. F.: Comparing F region ionospheric irregularity observations from C/NOFS and Jicamarca, Journal of Geophysical Research, 36, L00C01, https://doi.org/10.1029/2009GL038983, 2009.

Kelley, M.: The Earth's Ionosphere: Plasma Physics and Electrodynamics, International Geophysics, Elsevier Science, https://books.google.co.ug/books?id=3GlWQnjBQNgC, 2009.

Kelley, M. C., Rodrigues, F. S., Makela, J. J., Tsunoda, R., Roddy, P. A., Hunton, D. E., Retterer, J. M., de La Beaujardiere, O., de Paula, E. R., and Ilma, R. R.: C/NOFS and radar observations during a convective ionospheric storm event over South America, Geophysical Research Letters, 36, L00C07, https://doi.org/10.1029/2009GL039378, 2009.

Kil, H. and Heelis, R. A.: Global distribution of density irregularities in the equatorial ionosphere, Journal of Geophysical Research, 103, 407–418, https://doi.org/10.1029/97JA02698, 1998.

Nishioka, M., Saito, A., and Tsugawa, T.: Occurrence characteristics of plasma bubble derived from global ground-based GPS receiver networks, Journal of Geophysical Research: Space Physics, 113, https://doi.org/10.1029/2007ja012605, 2008.

Nishioka, M., Basu, S., Basu, S., Valladares, C. E., Sheehan, R. E., Roddy, P. A., and Groves, K. M.: C/NOFS satellite observations of equatorial ionospheric plasma structures supported by multiple ground-based diagnostics in October 2008, Journal of Geophysical Research: Space Physics, 116, n/a–n/a, https://doi.org/10.1029/2011ja016446, 2011.

Ossakow, S. L.: Ionospheric irregularities, Reviews of Geophysics, 17, 521–533, https://doi.org/10.1029/RG017i004p00521, 1979.

Roddy, P. A., Hunton, D. E., Ballenthin, J. O., and Groves, K. M.: Correlation of in situ measurements of plasma irregularities with ground-based scintillation observations, Journal of Geophysical Research (Space Physics), 115, A06303, https://doi.org/10.1029/2010JA015288, 2010.

Siefring, C. L., Bernhardt, P. A., Roddy, P. A., and Hunton, D. E.: Comparisons of equatorial irregularities measurements from C/NOFS: TEC using CERTO and CITRIS with in-situ plasma density, Geophysical Research Letters, 36, L00C08, https://doi.org/10.1029/2009GL038985, 2009.

Wang, Z., Shi, J. K., Torkar, K., Wang, G. J., and Wang, X.: Correlation between ionospheric strong range spread F and scintillations observed in Vanimo station, Journal of Geophysical Research: Space Physics, 119, 8578–8585, https://doi.org/10.1002/2014ja020447, 2014.

Zakharenkova, I., Astafyeva, E., and Cherniak, I.: GPS and in situ Swarm observations of the equatorial plasma density irregularities in the topside ionosphere, Earth, Planets, and Space, 68, 120, https://doi.org/10.1186/s40623-016-0490-5, 2016.

---

## Referee Comment (RC2) · Anonymous Referee #2 · 17 Feb 2020

General comment:

The authors compare density irregularities observed in-situ by the Swarm satellites with ground-based observations of plumes made by the Jicamarca unattended long term investigations of the ionosphere and atmosphere (JULIA) radar and ionosondes. Using data between 2014 and 2018, the authors investigate whether Swarm can be used as indicator of plasma plumes/Spread F observed on the ground. Showing a few case studies and by comparing the statistical trends between the in-situ and ground-based instruments, the authors conclude that Swarm can be used to detect the presence of well-developed plumes. The manuscript is carefully organized, the figures and presentation are clear, and the text is generally well written. While comparisons between in-situ and ground-based is worthwhile, and the datasets well suited, the novelty and importance of the findings are not significant enough, in my opinion. However, I think this work has potential values after significant additional work and resubmission.

Suggestions and comments are given below.

Major comments:

1. The author mention that their main focus is to determine whether Swarm can be used to detect plumes and Spread F. Even though previous studies used lower sampling resolution/single cases, I have difficulties understanding why the previous work(s) is(are) not sufficient to determine whether Swarm can detect plumes or not? There is even a standard Swarm data product called the Ionospheric Bubble Index (IBI) to detect equatorial irregularities (which does not seem to be referred to).

2. I agree that statistical studies can provide additional information. However, it is not clear to me what the authors actually get from their statistics, except for similar trends as the ones published in the literature (both from ground-based and Swarm "irregularity" studies). I am fairly certain that with their interesting dataset, the authors can reach more substantial conclusions, especially with the spatial/altitudinal separation and sampling rate of Swarm.

3. (section 3.3) The description of the method to calculate the statistics (selection of events, number of events, how occurrence rates are calculated, exact months, description of quantities shown in figures etc. ) is insufficient.

Minor comments: P=page, l=line

P.3 l.30: Doesn't the faceplate actually record Ni, from which Ne is calculated?

P.4 l.18: "vs" should be "versus"

P.4 l.29: "Ngwira et al. (2013)" should have (. . .)

[Figure]

P.4 l.30: Is the standard deviation also calculated on a 2sec running window? Are the scales selected important for you results?

P.5 l.7: "Smith et. Al (2015); Zhan et al. (2018)" should have (. . .)

P.5 l.12: Definition of ISR.

P.5 l.14: How exactly do the authors define "nighttime"?

P.5 l.19: Space between "explorer" and (Galkin. . .)

P.5 l.28: Definition of "QLat".

P.5 l28: "Much higher altitude". This is very subjective.

P.7 Figure 3. This figure and the explanation are not clear: do the authors bin the data by day of the month, and take the maximum for each day over 4 years?.

P.8 l.14: Definition of "EIA".

P.8 l 30: "20.05 LT to 21:00 LT" I don't read the same time interval from the Figure.

P.15-16 Figure 10 and Figure 11: Could the authors describe in more detail how those figures are obtained?

P.16 l.2: Didn't the authors use the 16 Hz? (Why is 2 Hz relevant here?)

P.17. l.1-3: The purpose of these sentences is unclear to me. It is well known that equatorial irregularities can cause scintillations.

P.17 l.12: "directory" – "directly"?

---

## Author Comment (AC2) · 11 Mar 2020

As authors of the manuscript angeo-2019-153, we thank the anonymous referee for the constructive suggestions and comments. In enhancing the quality of the paper, all the remarks we received on this research will be taken into consideration and we present our response to each of them individually below. For the convenience of the referee we have repeated in the response the relevant comments and then given texts we intend to add in the revised manuscript in blue.

**Response to Anonymous Referee #2 comments**

**General Comment:** The authors compare density irregularities observed in-situ by the Swarm satellites with ground-based observations of plumes made by the Jicamarca unattended long term investigations of the ionosphere and atmosphere (JULIA) radar and ionosondes. Using data between 2014 and 2018, the authors investigate whether Swarm can be used as indicator of plasma plumes/Spread F observed on the ground. Showing a few case studies and by comparing the statistical trends between the in-situ and ground-based instruments, the authors conclude that Swarm can be used to detect the presence of well-developed plumes. The manuscript is carefully organized, the figures and presentation are clear, and the text is generally well written. While comparisons between in-situ and ground-based is worthwhile, and the datasets well suited, the novelty and importance of the findings are not significant enough, in my opinion. However, I think this work has potential values after significant additional work and resubmission. Suggestions and comments are given below.

**Major Comments:**

1. The author mention that their main focus is to determine whether Swarm can be used to detect plumes and Spread F. Even though previous studies used lower sampling resolution/single cases, I have difficulties understanding why the previous work(s) is(are) not sufficient to determine whether Swarm can detect plumes or not? There is even a standard Swarm data product called the Ionospheric Bubble Index (IBI) to detect equatorial irregularities (which does not seem to be referred to).

**Response:** The differences between this work and the previous works may not have been demonstrated and highlighted clearly in our manuscript. In the revised version, we are going to improve our article organization and put more description to highlight this. Below are the points we want to address in this study:

- Most previous studies (e.g, Kelley et al., 2009; Siefring et al., 2009; Hysell et al., 2009; Roddy et al., 2010; Nishioka et al., 2011) have mostly compared zonally oriented in situ plasma density measurements from Communication Navigation Outage Forecasting System (C/NOFS) satellite with JULIA observations. The Swarm satellites revisit neatly the same area in orbits oriented in the merdional direction. Therefore, our study compares sub-kilometer in situ ionospheric irregularities recorded by Swarm in the meridional direction with observations from Jicamarca. We found that the results based on the JULIA radar and ionosonde agreed with the plasma density obtained from measurements of the Swarm faceplate for single satellite passes over or near the JRO.

- Previous comparison of Swarm in situ measurements with ground-based radar observations (Zakharenkova et al., 2016, e.g,) mostly used LP measurements at 2 Hz frequency. The faceplate carried by Swarm as part of the Electric Field Instrument (EFI) has enabled the discovery of small-scale (down to 500 km length scale along the spacecraft track) ionospheric irregularities. In this study, we used Swarm faceplate measurements at a frequency of 16 Hz. Coherent scatter radars e.g, the JULIA radar can monitor irregularities at high spatial resolution (3 m scale length for the case of JULIA) and therefore these were compared with Swarm faceplate observations of ionospheric irregularities of small scales. The high-resolution faceplate data enabled smaller scale structures to be identified in electron density. Also, previous comparison of Swarm in situ measurements with ground-based radar observations (Zakharenkova et al., 2016, e.g,) were mostly single case presentations. Our study provides an extended statistical analysis covering years from 2014 to 2018.

- As far as we know, a quantitative statistical relationship between plasma bubbles observed in situ in the meridional direction, 250 MHz amplitude scintillation, and JULIA observations were reported by Burke et al. (2003) using data recorded by the polar-orbiting Defense Meteorological Satellite Program (DMSP). However, DMSP orbited at an altitude of about 840 km and this was a limitation in that most ionospheric irregularities didnot ascend to DMSP altitude. In our study, we used the polar orbiting Swarm satellites which has provided a renewed opportunity to compare in situ and JULIA observation at altitudes of 460 km (Swarm A and C) and 510 km (Swarm B). Compared to DMSP, Swarm allows comparison of measurements from identical instruments at different altitudes and in different longitudinal sectors.

– Concerning the IBI index, it is a standard Level 2 product of the Swarm mission (Park et al., 2013). IBI provides information on climatology of ionospheric Irregularities itself as well as on the disturbance level of the magnetic field data by taking both electron density and magnetic field measurements into account. Following the referees suggestion, we accessed the IBI index for all the Swarm satellites from

`https://swarm-diss.eo.esa.int/#swarm%2FLevel2daily%2FLatest_baselines%2FIBI` and generated the distribution of ionospheric irregularities derived from the IBI as a function of quasi-dipole latitude and Local time. The results for Swarm C are presented here in Fig. 1. It is important to note that the IBI is just

[Figure]

**Figure 1.** Distribution of ionospheric irregularities derived from the IBI as a function of quasi-dipole latitude and Local time for Swarm C.

either 1, 0, -1 for bubble detected, not detected or undetermined, respectively. Therefore, to generate the results in the figure, the datasets were first grouped into different seasons corresponding to March Equinox (Feb-Mar-Apr), June Solstice (May-Jun-Jul), September Equinox (Aug-Sep-Oct), and December Solstice (Nov-Dec-Jan). For each season, the data was then binned into $1° \times 0.1\,\mathrm{hr}$ quasi-dipole latitude-local time bin. For each quasi-dipole latitude-local time bin, the percentage occurrence was obtained by dividing the number of observations with $\mathrm{IBI} = 1$ by the total number of observations. The distribution of ionospheric irregularities derived from the IBI shows similar seasonal dependence to the ones shown in Figure 11(see manuscript), with highest percentage occurrence in the equinoxes and December solstice.

– It is interesting to know how ionospheric irregularities are classified by the plasma bubbles detection algorithm. Therefore, we are going to consider adding the IBI index analysis in the revised version of the manuscript.

2. I agree that statistical studies can provide additional information. However, it is not clear to me what the authors actually get from their statistics, except for similar trends as the ones published in the literature (both from ground-based and Swarm "irregularity" studies). I am fairly certain that with their interesting dataset, the authors can reach more substantial conclusions, especially with the spatial/altitudinal separation and sampling rate of Swarm.

**Response:** After presenting single case examples in Fig. 4, 5, and 6, we wanted to investigate the characterisitics of the small-scale structures in more details with the help of a statistical study. Concerning the observations and conclusions from the statistical analysis, these may not have been demonstrated and highlighted clearly in our manuscript.

– Specifically for Fig. 7, the effect of spatial and altitudinal separation among the Swarm satellites can be observed and the following observations and conclusions will be added in the revised manuscript in this regard:

  • For instances when Swarm registered events, while JULIA and ionosonde recorded no signatures, we checked on the longitudinal separation between the satellite passes and the ground-site. The longitudinal separations obtained between the Swarm passes and the ground site were often $\approx 5°$ and the magnitude of the in situ perturbations were relatively low. Ionospheric irregularities tend to be magnetic field aligned (Ossakow, 1979; Kil and Heelis, 1998; Nishioka et al., 2008; Kelley, 2009) and therefore, Swarm may encounter irregularities in situ of relatively low magnitudes, while JULIA and ionosonde do not identify any events, for wider longitudinal offset of a pass from the ground site.

  • In addition, for irregularities observed by Swarm only, Swarm B had the lowest percentage occurrence close to about 5% compared to Swarm A/C. The observed difference may be because of the progressive temporal and altitudinal separation between Swarm B and A/C (Zakharenkova et al., 2016). Swarm B orbits at a higher altitude compared to A/C and it crosses the same region later than A/C.

  • Generally, in Fig. 7, a difference in percentage occurrence in all categories is observed between Swarm A and C although they orbit at the same altitude above sea-level. The large scale longitudinal bubble structure is sometimes observed with the two Swarm satellites (Xiong et al., 2016), but for small scale irregularities, the 1.5° longitudinal separation between the satellites is too large for a significant correlation between them.

The points stated above will be included in the revised manuscript.

3. (section 3.3) The description of the method to calculate the statistics (selection of events, number of events, how occurrence rates are calculated, exact months, description of quantities shown in figures etc. ) is insufficient.

**Response:** The description of the method to calculate the statistics may not have been described in detail. In the revised manuscript, we are going to incorporate detailed description of the methods we adopted. Below is the methods we used in the statistical analysis:

– For Section 3.3.1

- The Swarm satellites regress in longitude by about 15° between orbital ascending nodes. Therefore, in comparison with JULIA and ionosonde data, the Swarm passes were allowed to be within ±5° magnetic longitude of the JRO to make sure that a sufficient amount of Swarm passes could be used for the statistical examination. Summary plots such as those presented in Fig. 4 were generated for all days during the years from 2014 to 2018 for which all data sets existed. In total, 560 night-time orbits were used for which JULIA, Swarm, and ionosonde data were available concurrently. The outputs of the summary plots could be categorized into four cases considering the presence (or not) of irregularities. In general, these four cases are, "Irregularities observed both on the ground and in situ", 'No irregularities observed both on the ground and in situ", "Irregularities observed only in situ", and "Irregularities observed only on the ground". For each RTI plot, the SNR corresponding to the peak height was determined and an event was identified as a significant irregularity when the peak height was greater than or equal to 400 km. For peak height less than 400 km, these were classified as weak or no irregularities. For the in situ Swarm observations, we assigned a threshold of $1 \times 10^{10}$ m$^{-3}$ for std($\Delta N_e$) to be considered a significant irregularity event, while for std($\Delta N_e$) less than the threshold were considered as weak or no irregularities. For the ionosonde measurements, QF values greater than or equal to 20 km were considered as significant irregularity events.

- For each category, the percentage occurrence was computed as the ratio of the total number events in that category to the number of observations.

– For Section 3.3.2:

- To obtain the results presented in Fig. 8, ground-based JULIA SNR data for the years from 2014 to 2018 were used. To eliminate the impact of geomagnetically disturbed conditions on the statistical outcomes, the data were filtered and only those recorded during quiet geomagnetic conditions (Kp $\leq$ 3) were taken into account. (see Pg. 12 L. 7-8) The JULIA data accumulated for the years from 2014 to 2018 were sufficient for examining the seasonal variation. Therefore, the seasonal dependence of local time distribution of JULIA observations of ionospheric irregularities was also examined by grouping all the data into different seasons corresponding to March Equinox (Feb-Mar-Apr), June Solstice (May-Jun-Jul), September Equinox (Aug-

Sep-Oct), and December Solstice (Nov-Dec-Jan). For each local time-height bin, the percentage occurrence was obtained by dividing the number of observations with $\text{SNR} > 10$ dB by the total number of observations (Smith et al., 2016). (see Pg. 12 L. 9-10)

- Figure 10 shows the QF indices derived from ionosonde observations as a function of local time and months. To obtain the results presented in Fig. 10, ground-based ionosonde data for the years from 2014 to 2018 were also used. The data were also filtered and only those recorded during quiet geomagnetic conditions were considered. To generate the results presented in Figure 10, for each month (y-axis), the QF indices were averaged over 0.1 hr Local time bins.

- Figure 11 shows the QLat-LT distributions of $\text{std}(\Delta N_e)$ for the Swarm satellites for the years from 2014 to 2018. Recall that the Swarm passes were allowed to be within $\pm 5°$ magnetic longitude and $\pm 40°$ QLat. The results presented in Fig. 11 were also generated considering only the geomagnetically quiet conditions $(\text{Kp} \leq 3)$. The $N_e$ data collected for the five years were also grouped into different seasons similar to those presented in Fig. 9. The $\text{std}(\Delta N_e)$ was then calculated in bins of $1° \times 0.1$ hr resolution in QLat and Local time. The occurrence rate of ionospheric irregularities does not always correspond to the highest amplitude of irregularity structures (Wan et al., 2018). Therefore, we presented the calculated $\text{std}(\Delta N_e)$ per bin as a function of QLat and Local time as seen in Fig. 11.

  – In the revised manuscript, we are going to highlight statistical procedures as listed above.

**Minor Comments:**

P.3 l.30: Doesn't the faceplate actually record Ni, from which Ne is calculated?

**Response:** Ion density, $N_i$ is derived from the faceplate current assuming that the current is carried by ions hitting the faceplate due to the orbital motion of the spacecraft. However, due to quasi-neutrality $N_i$ must be equal to the electron density $N_e$ (Buchert, 2016).

This will be added in the manuscript for clarification.

P.4 l.18: "vs" should be "versus"

**Response:** This will be changed as suggested by the referee.

P.4 l.29: "Ngwira et al. (2013)" should have (. . .)

**Response:** The citation identified by the referee will be changed as suggested.

P.4 l.30: Is the standard deviation also calculated on a 2sec running window? Are the scales selected important for you results?

Response: The standard deviation of the residuals was calculated at a running window of 2-s and this was used to represent the magnitude of the electron density perturbation.

This information will be added in the revised manuscript.

Concerning the scales selected, it is important for our results since we concentrated on small-scale ionospheric irregularities. The window to calculate $dN_e$ and the standard deviation (for the case of small-scale ionospheric irregularities) should be short, but has to be long enough to avoid spurious detection of ionospheric irregularities. We also tried 1 s, 16 points, instead of 2s, 32 points, and the outcome was similar and reasonable.

In addition, the Coherent scatter radars e.g, the JULIA radar can monitor irregularities at high spatial resolution (3 m scale length for the case of JULIA) and therefore these were compared with Swarm faceplate observations of ionospheric irregularities of small scales which were quantified by using a running window of 2-s.

P.5 l.7: "Smith et. Al (2015); Zhan et al. (2018)" should have (. . .)

Response: The citations identified by the referee will be changed as suggested.

P.5 l.12: Definition of ISR.

Response: ISR stands for Incoherent Scatter Radar. This will be added in the revised manuscript.

P.5 l.14: How exactly do the authors define "nighttime"?

Response: In our manuscript, nighttime is defined as the time period between 1800 LT and 0600 LT. The sentence in P.5 L. 14 will be rephrased in the revised manuscript to make this clear as follows:

Therefore, to compare the Swarm observations with the JULIA measurements, only swarm satellite passes for the time period between 1800 LT and 0600 LT were considered.

P.5 l.19: Space between "explorer" and (Galkin. . .)

Response: This will be adjusted as suggested by the referee.

P.5 l.28: Definition of "QLat".

Response: QLat stands for quasi-dipole latitude (Laundal and Richmond, 2016). This will be added in the revised manuscript.

P.5 l28: "Much higher altitude". This is very subjective.

**Response:** The identified words will be removed from the revised manuscript and the sentence will be rephrased to:

Also, Swarm B which orbited at about 510 km altitude above sea level recorded ionospheric irregularity structures on 2015-04-05 as seen from the right panel of Fig. 1.

P.7 Figure 3. This figure and the explanation are not clear: do the authors bin the data by day of the month, and take the maximum for each day over 4 years?.

**Response:** The maximum ranges were obtained for each "day of the month" over the four years when the JULIA data was available. As identified by the referee, there indeed seems not be a direct relation between "Maximum range" and "Time (days)". Instead each maximum range corresponds to "a day of the month". Therefore, the xlabel of Figure 3 will be changed to "Day of the month".

P.8 l.14: Definition of "EIA".

**Response:** EIA stands for Equatorial Ionization Anomally. This will be added in the revised manuscript.

P.8 l 30: "20.05 LT to 21:00 LT" I don't read the same time interval from the Figure.

**Response:** To make it clearer, the x-axis tick labels for LT will be changed to LT(hh:mm) in the revised manuscript as shown here in Fig. 2: To be exact, the time interval will be changed to 20:27 LT to 21:05 LT, in the revised manuscript.

P.15-16 Figure 10 and Figure 11: Could the authors describe in more detail how those figures are obtained?

**Response:** The referee is referred to the response to major comment 3.

P.16 l.2: Didn't the authors use the 16 Hz? (Why is 2 Hz relevant here?)

**Response:** Since we concentrated on the 16 Hz $N_e$, in the revised manuscript, the sentence will be rephrased to:

- The Swarm electron density measurements used in this study are limited by the sampling rate of 16 Hz and the orbital velocity of about 7.5 km/s to wavelengths of about 500 m, respectively, and longer.

P.17. l.1-3: The purpose of these sentences is unclear to me. It is well known that equatorial irregularities can cause scintillations.

**Response:** The sentence will be removed in the revised manuscript.

P.17 l.12: "directory" – "directly"?

**Response:** This will be changed in the revised manuscript as suggested by the reviewer.

[revised manuscript text omitted]

---

## Author Response (AR1)

**Simultaneous Ground-based and In Situ Swarm Observations of Equatorial F-region Irregularities over Jicamarca**

by Aol et al.

Dear Dr. Keisuke Hosokawa,

Thank you for your letter. As authors of the manuscript angeo-2019-153, we thank the referees for their constructive suggestions and comments. In enhancing the quality of the paper, all the remarks we received on this research were taken into consideration and we present our response to each of them individually below. A marked-up manuscript version has also been embedded at the end of this document. We hope, our manuscript is acceptable for Annalese Geophysicae in this form.

Best regards

Sharon Aol
* * *
Below you find our point-by-point reply. For the convenience of the referees we have repeated in the response the relevant comments and then given our answers in blue text.

**Response to Anonymous Referee #1 comments**

**Comment:** Present work describes comparison of ionospheric irregularity data between the satellite SWARM electron densities and ground-based JULIA radar data and ionosonde data,using the data obtained over Peru from 2014 to 2018. In terms of identification of ionospheric irregularities, they found good similarities between the three different data set. Their final conclusion was that SWARM data can be used as a tool to indicate presence of plasma plumes and Spread F occurrence. The data sets used in the manuscript are interesting and comparison of them in terms of plasma bubble finding, or validation study for identification of the plasma irregularities, are useful. However, if one asks what is a new finding in this work I could not find anything. The authors could have a new aspect if they further discuss in the data analysis, I guess. Therefore my conclusion is that the present work will not be acceptable as a scientific paper without any further scientific new finding.

**Response:** We are glad that the referee finds the data sets used in our paper interesting and comparison of them in terms of plasma bubble finding, or validation study for identification of the plasma irregularities useful.

The main focus, findings and innovations may not have been demonstrated and highlighted clearly in our manuscript. In the revised version, we have improved on our article organization and put more description to highlight our findings. Below are the points we want to address in this study and these have been highlighted in the revised manuscript.Please see the detailed descriptions in the revised manuscript at the Line numbers indicated:

– The main focus of this study is to determine whether Swarm in situ observations can be used as indicators of the presence of plasma plumes and spread-F on the ground by comparing simultaneous observations of plasma plumes by the Jicamarca unattended long term investigations of the ionosphere and atmosphere (JULIA) radar, ionogram spread F generated from ionosonde observations installed at the Jicamarca Radio Observatory (JRO), and irregularities observed in situ by Swarm. The combined multi-instrument measurements provide a more integrated and comprehensive way to study the morphological structure, development, and seeding mechanism of ionospheric irregularities.

– It should be noted that although ionospheric irregularities have been studied extensively, uncertainties still exist in understanding their evolution because of their varying scale sizes (Abdu, 2001; Sripathi et al., 2008; Aa et al., 2020). In this regard, different instruments are limited to observing ionospheric irregularities of particular scale size (Sripathi et al., 2008; Aa et al., 2020). Therefore, coordinated observation of ionospheric irregularities using different instruments is an effective way to generate an integrated and comprehensive image for specifying ionospheric irregularities of different scale sizes (e.g, Sripathi et al., 2008; Cherniak et al., 2019; Aa et al., 2019, 2020).(see Pg.2 L.24-29)

– Most previous studies (e.g, Kelley et al., 2009; Siefring et al., 2009; Hysell et al., 2009; Roddy et al., 2010; Nishioka et al., 2011) have compared zonally oriented in situ plasma density measurements from Communication Navigation Outage Forecasting System (C/NOFS) satellite with JULIA observations. The Swarm satellites revisit neatly the same area in orbits oriented in the merdional direction. Therefore, our study compares sub-kilometer in situ ionospheric irregularities recorded by Swarm in the meridional direction with observations from Jicamarca. We found that the results based on the JULIA radar and ionosonde agreed with the plasma density obtained from measurements of the Swarm faceplate for single satellite passes over or near the JRO. (see Pg.3 L.1-7)

– Previous comparison of Swarm in situ measurements with ground-based radar observations (Zakharenkova et al., 2016, e.g,) mostly used LP measurements at 2 Hz frequency. The faceplate carried by Swarm as part of the Electric Field Instrument (EFI) has enabled the discovery of small-scale (down to 500 km length scale along the spacecraft track) ionospheric irregularities. In this study, we used Swarm faceplate measurements at a frequency of 16 Hz. Coherent scatter radars e.g, the JULIA radar can monitor irregularities at high spatial resolution (3 m scale length for the case of JULIA) and therefore these were compared with Swarm faceplate observations of ionospheric irregularities of small scales. The high-resolution faceplate data enabled smaller scale structures to be identified in electron density. Also, previous comparison of Swarm in situ measurements with ground-based radar observations (Zakharenkova et al., 2016, e.g,) were mostly single case presentations. Our study provides an extended statistical analysis covering years from 2014 to 2018.(see Pg.3 L.15-19)

– As far as we know, a quantitative statistical relationship between plasma bubbles observed in situ in the meridional direction, 250 MHz amplitude scintillation, and JULIA observations were reported by Burke et al. (2003) using data recorded by the polar-orbiting Defense Meteorological Satellite Program (DMSP). However, DMSP orbited at an altitude of about 840 km and this was a limitation in that most ionospheric irregularities didnot ascend to DMSP altitude. In our study, we used the polar orbiting Swarm satellites which has provided a renewed opportunity to compare in situ and JULIA observation at altitudes of 460 km (Swarm A and C) and 510 km (Swarm B). Compared to DMSP, Swarm allows comparison of measurements from identical instruments at different altitudes and in different longitudinal sectors. (see Pg. 3 L. 8-15)

– As far as we know, Wang et al. (2014) were among the first to make concurrent observations of strong range spread-F and ionospheric irregularities measured in situ using ROCSAT-1 satellite and they found that strong spread-F were caused by the ionospheric irregularities. However, ROCSAT-1 orbited at about 600 km altitude with 35° orbital inclination. Therefore, we also compared the JULIA and Swarm observations of ionospheric irregularities with spread-F signatures recorded by an ionosonde colocated with the JULIA radar.(see Pg. 3 L. 29-33)

**Minor Comments:**

**Comment1:** Page 4, line 29, "Ngwira et al. (2013a)": change to"(Ngwira et al., 2013a)

**Response:** The citation has been changed as suggested. (see Page 5 line 11)

**Comment2:** Page 5, line 8, "Smith et al,,,,": change to "(Smith et al., 2015;Zhan et al. 2018)

**Response:** The citation has been changed as suggested. (see Page 5 line 20)

**Comment3:** Page 5, line 29, "from the left panel of Fig. 1": change to "fromthe right panel of Fig. 1"

**Response:** The sentence has been changed as suggested by the referee. (see Page 6 line 6-7)

**Comment4:** Page 7, Figure 3: Please explain why the authors plotted themaximum ranges as a function of time (days). It seems to have no relation between themaximum range and Time(days).

**Response:** The maximum ranges were plotted to check the altitude coverage of the various types of plumes observed by the JULIA radar compared to the Swarm altitudes. The maximum ranges were obtained for each "day of the month" when the JULIA data was available. As identified by the referee, there indeed seems not be a direct relation between "Maximum range" and "Time (days)". Instead each maximum range corresponds to "a day of the month". Therefore, the x-label of Figure 3 has been changed to "Day of the month". (see Fig. 3 Pg. 7)

**Comment5:** Page 12, Figure 7: The authors did not discuss in the case of "Irregularities observed by SWARM only". According to the height coverage of Julia radar (90 to 800 km), any irregularities detected by SWARM should be detectedby Julia radar. It could be difference of threshold of detection amplitude (?), further discussion would be helpful for readers.

**Response:** Ideally, any irregularity detected by Swarm would be detected by JULIA depending on the proximity of the Swarm pass to the JULIA longitude and the magnitude of the irregularitiy. Given the longitudinal range ($\pm5°$) that was used for good statistical coverage (see Pg 10 Line 11), there were chances that Swarm would record

ionospheric irregularities without JULIA identifying any plume structures. This is because ionospheric irregularities tend to be magnetic field aligned (Ossakow, 1979; Kil and Heelis, 1998; Nishioka et al., 2008; Kelley, 2009). An example is presented here in Figure 1. On the day 2015-11-08, Swarm encountered ionospheric irregularities,

[Figure]

**Figure 1.** Example of collocated observation by Swarm and JULIA on 2015-11-08.

while JULIA recorded no plume structures. Clearly, the Swarm passes were offset from the JULIA longitude to the east. The following discussion has been added in the revised manuscript, to explain "Irregularities observed by SWARM only"

For instances when Swarm registered events, while JULIA and ionosonde recorded no signatures, we checked on the longitudinal separation between the satellite passes and the ground-site. The longitudinal separations obtained between the Swarm passes and the ground site were often $\approx 5°$ and the magnitude of the in situ perturbations were relatively low. Ionospheric irregularities tend to be magnetic field aligned (Ossakow, 1979; Kil and Heelis, 1998; Nishioka et al., 2008; Kelley, 2009) and therefore, Swarm may encounter irregularities in situ of relatively low magnitudes, while JULIA and ionosonde do not identify any events, for wider longitudinal offset of a pass from the ground site.(see Pg.11 L.19 -24)

**Comment6:** Page 16, line 1, "The Bragg condition": Please explain what is "Bragg condition" for readers who are not familiar to the phrase.

**Response:** The Bragg condition explains the effect of periodic or quasi-periodic variations of the refractive index on the propagation of electromagnetic waves when the scales of the variations are of the order of the wavelength. The radio waves fulfilling the Bragg condition scatter at periodic variations of the refractive index such that a coherent superposition of reflections occurs and a reflected waves propagates in the direction of the radar receiver. The Bragg condition describes when the constructive interference occurs in a certain ratio between the transmitted wavelength and the distance of the reflective sub-surfaces:

$$d = \frac{\lambda_t}{2.\cos\theta}$$

$d$=distance between maxima and minima of the refractive index
$\lambda_t$=transmitted wavelength
$\theta$=incident angle

The following sentence has been included in the manuscript as further explanation of the "Bragg condition":

The Bragg condition for backscatter means that a radar can only observe structures in the refractive index with scale sizes close to the half radar wavelength (Kelley, 2009; Hocking et al., 2016).(see Pg. 17 L. 2-3)

**Comment7:** Page17, Conclusions, lines 13-14: "A few exceptions were also observed when,,,": Furtherscientific discussion would be valuable for readers

**Response:** This has been addressed in response to **Comment5**.

In addition, the following sentences have been included in the conclusion as further discussion for the observation made in lines 13-14 of the previous version of the manuscript:

For the case when JULIA and ionosonde recorded irregularity signatures, while Swarm observed no structures, the plume structures may not have ascended to Swarm altitudes by the time the satellites passed over Jicamarca or the satellites were simply in a different location. Swarm was able to detect ionospheric irregularities in situ, while no signature was recorded on the ground because the irregularities occurred at magnetic longitudes which were largely offset from the longitude of the ground-site.(see Pg. 17 L. 27-32)

**Response to Anonymous Referee #2 comments**

**General Comment:** The authors compare density irregularities observed in-situ by the Swarm satellites with ground-based observations of plumes made by the Jicamarca unattended long term investigations of the ionosphere and atmosphere (JULIA) radar and ionosondes. Using data between 2014 and 2018, the authors investigate whether Swarm can be used as indicator of plasma plumes/Spread F observed on the ground. Showing a few case studies and by comparing the statistical trends between the in-situ and ground-based instruments, the authors conclude that Swarm can be used to detect the presence of well-developed plumes. The manuscript is carefully organized, the figures and presentation are clear, and the text is generally well written. While comparisons between in-situ and ground-based is worthwhile, and the datasets well suited, the novelty and importance of the findings are not significant enough, in my opinion. However, I think this work has potential values after significant additional work and resubmission. Suggestions and comments are given below.

**Major Comments:**

**Comment 1:** The author mention that their main focus is to determine whether Swarm can be used to detect plumes and Spread F. Even though previous studies used lower sampling resolution/single cases, I have difficulties understanding why the previous work(s) is(are) not sufficient to determine whether Swarm can detect plumes or not? There is even a standard Swarm data product called the Ionospheric Bubble Index (IBI) to detect equatorial irregularities (which does not seem to be referred to).

**Response:** The differences between this work and the previous works may not have been demonstrated and highlighted clearly in our manuscript. In the revised version, we have put more description to highlight this. Below are the points we want to address in this study:

- Most previous studies (e.g, Kelley et al., 2009; Siefring et al., 2009; Hysell et al., 2009; Roddy et al., 2010; Nishioka et al., 2011) have compared zonally oriented in situ plasma density measurements from Communication Navigation Outage Forecasting System (C/NOFS) satellite with JULIA observations. The Swarm satellites revisit neatly the same area in orbits oriented in the merdional direction. Therefore, our study compares sub-kilometer in situ ionospheric irregularities recorded by Swarm in the meridional direction with observations from Jicamarca.(see Pg.3 L. 2-7)

- Previous comparison of Swarm in situ measurements with ground-based radar observations mostly used LP measurements at 2 Hz frequency (e.g, Zakharenkova et al., 2016). In this study, we used Swarm faceplate measurements at a frequency of 16 Hz. The faceplate carried by Swarm as part of the Electric Field Instrument (EFI) has enabled the discovery of small-scale (down to 500 km length scale along the spacecraft track) ionospheric irregularities. Coherent scatter radars e.g, the JULIA radar can monitor irregularities at high spatial resolution (3 m scale length for the case of JULIA) and therefore these were compared with Swarm faceplate observations of ionospheric irregularities of small scales. The high-resolution faceplate data enabled smaller scale structures to be identified in electron density. Also, previous comparison of Swarm in situ measurements with ground-based radar observations (e.g, Zakharenkova et al., 2016) were mostly single case presentations. Our study provides an extended statistical analysis covering years from 2014 to 2018.(see Pg. 3 L. 15-19)

- As far as we know, a quantitative statistical relationship between plasma bubbles observed in situ in the meridional direction, 250 MHz amplitude scintillation, and JULIA observations were reported by Burke et al. (2003) using data recorded by the polar-orbiting Defense Meteorological Satellite Program (DMSP). However, DMSP orbited at an altitude of about 840 km and this was a limitation in that most ionospheric irregularities didnot ascend to DMSP altitude. In our study, we used the polar orbiting Swarm satellites which has provided a renewed opportunity to compare in situ and JULIA observation at altitudes of 460 km (Swarm A and C) and 510 km (Swarm B). Compared to DMSP, Swarm allows comparison of measurements from identical instruments at different altitudes and in different longitudinal sectors. (see Pg. 3 L. 8-15)

  – Concerning the IBI index, it is a standard Level 2 product of the Swarm mission (Park et al., 2013). IBI provides information on climatology of ionospheric irregularities itself as well as on the disturbance level of the magnetic

field data by taking both electron density and magnetic field measurements into account. Following the referees suggestion, we accessed the IBI index for all the Swarm satellites from `https://swarm-diss.eo.esa.int/#swarm%2FLevel2daily%2FLatest_baselines%2FIBI` and generated the distribution of ionospheric irregularities derived from the IBI as a function of quasi-dipole latitude and Local time. The results for Swarm C are presented here in Fig. 2. It is important to note that the IBI is just

[Figure]

**Figure 2.** Distribution of ionospheric irregularities derived from the IBI as a function of quasi-dipole latitude and Local time for Swarm C.

either 1, 0, -1 for bubble detected, not detected or undetermined, respectively. Therefore, to generate the results in the figure, the datasets were first grouped into different seasons corresponding to March Equinox (Feb-Mar-Apr), June Solstice (May-Jun-Jul), September Equinox (Aug-Sep-Oct), and December Solstice (Nov-Dec-Jan). For each season, the data was then binned into $1° \times 0.1\,\mathrm{hr}$ quasi-dipole latitude-local time bin. For each quasi-dipole latitude-local time bin, the percentage occurrence was obtained by dividing the number of observations with $\mathrm{IBI} = 1$ by the total number of observations. The distribution of ionospheric irregularities derived from the IBI shows similar seasonal dependence to the ones shown in Figure 11(see manuscript), with highest percentage occurrence in the equinoxes and December solstice. The latitude-LT plots of IBI do not offer as much details as the density measurements. However, the comparison reveals also differences. While the absolute irregularity strength largely shows the familiar double peaks at roughly $\pm15°$ north and south of the magnetic equator, the IBI has almost no clear double structure and reflects only periods with very strong irregularities. The reliability of IBI as an indicator of Spread F and scintillations of radio signals seems doubtful to us.

– It is interesting to know how ionospheric irregularities are classified by the plasma bubbles detection algorithm. Therefore, we have added the IBI index analysis in the revised version of the manuscript. (see Pg 16. L.3-15 and Figure. 12)

**Comment 2:** I agree that statistical studies can provide additional information. However, it is not clear to me what the authors actually get from their statistics, except for similar trends as the ones published in the literature (both from ground-based and Swarm "irregularity" studies). I am fairly certain that with their interesting dataset, the authors can reach more substantial conclusions, especially with the spatial/altitudinal separation and sampling rate of Swarm.

**Response:** Concerning the observations and conclusions from the statistical analysis, these may not have been demonstrated and highlighted clearly in our manuscript.

– Specifically for Fig. 7, the effect of spatial and altitudinal separation among the Swarm satellites can be observed and the following observations and conclusions have been added in the revised manuscript in this regard:

- For instances when Swarm registered events, while JULIA and ionosonde recorded no signatures, we checked on the longitudinal separation between the satellite passes and the ground-site. The longitudinal separations obtained between the Swarm passes and the ground site were often $\approx 5°$ and the magnitude of the in situ perturbations were relatively low. Ionospheric irregularities tend to be magnetic field aligned (Ossakow, 1979; Kil and Heelis, 1998; Nishioka et al., 2008; Kelley, 2009) and therefore, Swarm may encounter irregularities in situ of relatively low magnitudes, while JULIA and ionosonde do not identify any events, for wider longitudinal offset of a pass from the ground site.(see Pg. 11 L. 19-25)

- In addition, for irregularities observed by Swarm only, Swarm B had the lowest percentage occurrence close to about 5% compared to Swarm A/C. The observed difference may be because of the progressive temporal and altitudinal separation between Swarm B and A/C (Zakharenkova et al., 2016). Swarm B orbits at a higher altitude compared to A/C and it crosses the same region later than A/C. Generally, in Fig. 7, a difference in percentage occurrence in all categories is observed between Swarm A and C although they orbit at the same altitude above sea-level. The large scale longitudinal bubble structure is sometimes observed with the two Swarm satellites (Xiong et al., 2016), but for small scale irregularities, the 1.5° longitudinal separation between the satellites is too large for a significant correlation between them.(see Pg. 12 L. 1-7)

**Comment 3:** (section 3.3) The description of the method to calculate the statistics (selection of events, number of events, how occurrence rates are calculated, exact months, description of quantities shown in figures etc. ) is insufficient.

**Response:** The description of the method to calculate the statistics may not have been described in detail. In the revised manuscript, we have incorporated detailed description of the methods we adopted. Below is the methods we used in the statistical analysis:

– For Section 3.3.1

[revised manuscript text omitted]

**Minor Comments:**

P.3 l.30: Doesn't the faceplate actually record Ni, from which Ne is calculated?

**Response:** Ion density, $N_i$ is derived from the faceplate current assuming that the current is carried by ions hitting the faceplate due to the orbital motion of the spacecraft. However, due to quasi-neutrality $N_i$ must be equal to the electron density $N_e$ (Buchert, 2016).

This has been added in the revised manuscript for clarification. (see Pg. 4 L. 10-13 )

P.4 l.18: "vs" should be "versus"

**Response:** This has been changed as suggested. (see Pg. 4 L. 33 )

P.4 l.29: "Ngwira et al. (2013)" should have (. . .)

**Response:** The citation identified has been changed as suggested. (see Pg. 5 L. 11)

P.4 l.30: Is the standard deviation also calculated on a 2sec running window? Are the scales selected important for you results?

**Response:** The standard deviation of the residuals was calculated at a running window of 2-s and this was used to represent the magnitude of the electron density perturbation.

This information has been added in the revised manuscript.(see Pg. 5 L. 12)

Concerning the scales selected, it is important for our results since we concentrated on small-scale ionospheric irregularities. The window to calculate $dN_e$ and the standard deviation (for the case of small-scale ionospheric irregularities) should be short, but long enough to avoid spurious detection of ionospheric irregularities. We also tried 1 s, 16 points, instead of 2s, 32 points, and the outcome was similar and reasonable.

In addition, the Coherent scatter radars e.g, the JULIA radar can monitor irregularities at high spatial resolution (3 m scale length for the case of JULIA) and therefore these were compared with Swarm faceplate observations of ionospheric irregularities of small scales which were quantified by using a running window of 2-s.

P.5 l.7: "Smith et. Al (2015); Zhan et al. (2018)" should have (. . .)

**Response:** The citations identified have been changed as suggested. (see Pg. 5 L 20)

P.5 l.12: Definition of ISR.

**Response:** ISR stands for Incoherent Scatter Radar. This has been added in the revised manuscript. (see Pg. 5 L 24)

P.5 l.14:    How exactly do the authors define "nighttime"?

**Response:** In our manuscript, nighttime is defined as the time period between 1800 LT and 0600 LT. The sentence has been rephrased in the revised manuscript to make this clear as follows:

Therefore, to compare the Swarm observations with the JULIA measurements, only swarm satellite passes for the time period between 1800 LT and 0600 LT were considered. (see Pg. 5 L 27)

P.5 l.19:    Space between "explorer" and (Galkin. . .)

**Response:** This has been adjusted as suggested. (see Pg. 5 L 32)

P.5 l.28:    Definition of "QLat".

**Response:** QLat stands for quasi-dipole latitude (Laundal and Richmond, 2016). This has been added in the revised manuscript. (see Pg. 6 L 5)

P.5 l28:    "Much higher altitude". This is very subjective.

**Response:** The identified words have been removed from the revised manuscript and the sentence has been rephrased to:

Also, Swarm B which orbited at about 510 km altitude above sea level recorded ionospheric irregularity structures on 2015-04-05 as seen from the right panel of Fig. 1.(see Pg. 6 L 6-7)

P.7 Figure 3.    This figure and the explanation are not clear: do the authors bin the data by day of the month, and take the maximum for each day over 4 years?.

**Response:** The maximum ranges were obtained for each "day of the month" over the four years when the JULIA data was available. As identified by the referee, there indeed seems not to be a direct relation between "Maximum range" and "Time (days)". Instead each maximum range corresponds to "a day of the month". Therefore, the x-label of Figure 3 has been changed to "Day of the month". (see Fig. 3 in the revised manuscript)

P.8 l.14:    Definition of "EIA".

**Response:** EIA stands for Equatorial Ionization Anomaly. This has been added in the revised manuscript. (see Pg.8 L 11)

P.8 l 30:    "20.05 LT to 21:00 LT" I don't read the same time interval from the Figure.

**Response:** To make it clearer, the x-axis tick labels for LT have been changed to LT(hh:mm) in the revised manuscript as shown here in Fig. 3: To be exact, the time interval have been changed to 20:27 LT to 21:05 LT, in the revised manuscript. (see Pg. 9 L. 16)

P.15-16    Figure 10 and Figure 11: Could the authors describe in more detail how those figures are obtained?

**Response:** This has been discussed in response to major comment 3.

P.16 l.2:    Didn't the authors use the 16 Hz? (Why is 2 Hz relevant here?)

**Response:** Since we concentrated on the 16 Hz $N_e$, in the revised manuscript, the sentence has been rephrased to:

- The Swarm electron density measurements used in this study are limited by the sampling rate of 16 Hz and the orbital velocity of about 7.5 km/s to wavelengths of about 500 m, respectively, and longer. (see Pg. 17 L. 3-5)

P.17. l.1-3:    The purpose of these sentences is unclear to me. It is well known that equatorial irregularities can cause scintillations.

**Response:** The sentence has been removed in the revised manuscript.

P.17 l.12:    "directory" – "directly"?

**Response:** This has been changed in the revised manuscript as suggested by the reviewer.(see Pg. 17 L. 24)

[revised manuscript text omitted]

---

## Author Response (AR2)

**Simultaneous Ground-based and In Situ Swarm Observations of Equatorial F-region Irregularities over Jicamarca**

by Aol et al.

Dear Dr. Keisuke Hosokawa,

Thank you for your letter. As authors of the manuscript angeo-2019-153, we thank the referees for their constructive suggestions and comments. We have implemented all the suggestions and comments given by the reviewers, and believe that this has greatly improved the quality of this paper. A marked-up manuscript version has also been embedded at the end of this document. We hope, our manuscript is now acceptable for Annalese Geophysicae in this form.

Best regards

Sharon Aol
* * *
Below you find our point-by-point reply. For the convenience of the referees we have repeated in the response the relevant comments and then given our answers in blue text.

**Response to Anonymous Referee #1 comments**

**Minor Comment:** 2nd Review of "Simultaneous Ground-based and In Situ Swarm Observations of Equatorial F-region Irregularities over Jicamarca". The authors attended most of my comments. The manuscript was improved significantly, except one point: Regarding my comment (4) for Figure 3, my question was that there is no (physical) meaning to show the maximum ranges (y-axis) as a function of day of the month (x-axis). Because, one day (of the month) includes from all of the months and years from 2014 to 2018, including all kind of temporal variabilities. If the authors want to show statistical trends of maximum ranges observed by JULIA groundbased data and to compare the altitude range of SWARM data, the graph could be Frequency of occurrence of the maximum range (y-axis) against vertical height range from 100 to 900 km (in the x-axis), and to mark (highlighted) the range of SWARM orbit. If the authors agree, they can revise the figure. End of Comment

**Response:** We are glad that the referee has found significant improvement on our revised manuscript and we thank the referee for accepting our manuscript subject to minor revisions.

Concerning Figure 3, the referees suggestion has been taken into consideration. Figure 3 has been replaced with a graph of Frequency of occurrence of the maximum range (y-axis) against maximum range (x-axis) for topside, bottom-side and bottom-type spread Fs, independently. We also present the result here: The gray region represents the altitude range of

the Swarm satellites. The Swarm altitude range coincides with high percentage occurrence of maximum range of topside plasma plumes.

The text within the revised manuscript in relation to Fig. 3 has been changed to:

– Figure 3 shows the frequency of occurrence of the maximum height achieved by the various types of plumes for the years from 2014 to 2018. From Fig. 3, the Swarm altitude range coincides with high frequency of occurrence of maximum range of topside plasma plumes. This reveals that the Swarm orbits are most suitable to detect topside plasma plumes compared to the bottom-side and bottom-type spread Fs (please see Pg. 6 L. 28 and Pg. 7 L. 1-4 )

The caption of Fig. 3 has also been changed in the revised manuscript to:

– Frequency of occurrence of the maximum height achieved by the different types of ESFs observed by JULIA radar for the years from 2014 to 2018. The gray region indicates the approximate altitude coverage of the Swarm satellites from 2014 to 2018. (please see caption of Fig. 3 in the revised manuscript)

**Response to Anonymous Referee #2 comments**

**General Comment:** The authors compare density irregularities observed in-situ by the Swarm satellites with ground-based observations of plumes made by the Jicamarca unattended long term investigations of the ionosphere and atmosphere (JULIA) radar and ionosondes. Using data between 2014 and 2018, the authors investigate whether Swarm can be used as indicator of plasma plumes/Spread F observed on the ground, and compare the statistical trends between the in-situ and ground-based instruments. This study provides a nice confirmation that Swarm can detect plumes and Spread F and, having conjunctions from different instruments grouped in one manuscript is of significant interest. However, even though the authors attempted to address the comments from the previous reviews, the main messages and discussion are still somehow unclear. This may also partially be due to the structure of the text where observations and discussions merged, making the global arguments less clear. Since the study is very similar to previous ones, my point of view is that the results should be discussed in more detail (especially with respect to the other, almost similar studies) and the main findings emphasized. Furthermore, some parts should be clarified.

Further details and comments are given below. (pages P and lines l refer to the final version of the manuscript)

**Response:** We are glad that the referee finds the study of significant interest and that the study provides a nice confirmation that Swarm can detect irregularities and plasma bubbles associated with plumes and spread F. Concerning the discussion of previous studies that addressed nearly similar concepts, these may not have been demonstrated clearly. In the revised manuscript, we have improved on our discussions with respect to previous studies in the "Results and Discussions" section of the manuscript and emphasized our main findings. Please see our responses to the comments below:

**Comment 1: Discussion** From what I understand, the authors argue that the main differences between their work and the previous ones are (very briefly summarized)
a) They study meridional structures (and that few have done this before, and DMSP was too high)
b) They used 16 Hz data (LP 2 Hz data was used before for Swarm).
c) Previous JULIA-Swarm comparison was mostly from a case study (using 2Hz)

While the "technical differences" are mentioned in the introduction (P3), there is no real substantial discussion on comparing the author's results with the previous studies. Since the studies are similar, some more detail discussion is expected to highlight differences/similarities in the results. E.g. of ideas of discussion points: Differences between studies based on zonal and merional? Comparison with results of DMSP? Is it actually necessary to use the 16 Hz or is 2 Hz (as done in the previous studies) sufficient? Since the scales detected by JULIA and Swarm are different anyways... etc

**Response:** The following discussions have been included in the revised manuscript to improve our discussions in the "Results and Discussions" section with respect to previous studies which have addressed almost similar concept and where we already discussed earlier studies have also been pointed out. (please note that the added discussions are in blue text below.)

[revised manuscript text omitted]

**Comment 2: Statistical analysis** While looking at the statistical analysis and now that the authors explained how they performed their analysis, I have a few questions:

a) Comparing Fig. 8, 9, and 10, it seems like the largest occurrence of plasma plumes about 400km (so the ones that can be detected by Swarm) occur around December solstice, which also corresponds to the period with peak in mean QF. The distribution of the std(Ne) from Swarm do not seem to exhibit this pattern (but maybe the actual occurrence rate of ionospheric irregularities detected by Swarm would?). Could the authors comment on this and on the eventual implications for using Swarm as indicator of Spread F.

**Response:** The largest occurrence of plasma plumes appear to occur in December solstice as viewed from Fig. 8, 9, and 10, while for the case of distribution of std($\Delta N_e$) in Figure 11 the March equinox seems to be the peak, with December solstice a close second. Following the referee's suggestion, we re-generated Figure 11 to represent percentage occurrence of ionospheric irregularities and nearly similar trend was observed (please see example results below for Swarm C). Therefore, we preferred to maintain the distribution of std ($\Delta N_e$) because the occurrence rate of ionospheric irregularities does not always correspond to the highest amplitude of irregularity structures (Wan et al., 2018). Slight

discrepancies may arise because of counting statistics given the fact that the orbital planes of Swarm drift progressively with time. Therefore, each season has a different count of passes available during the local time range of 1800 LT to 0600 LT. Despite the slight discrepancies because of counting statistics especially for Swarm, the overall seasonal trend is similar for std($\Delta N_e$), QF and plumes observed by JULIA.

– Further more, particularly in Figure 11 the colors emphasize rather anomaly crests (about ±15° QLat), while the ground-based data are from the magnetic equator. We prefer not to draw conclusions from the comparion regarding the seasons. The main advantage of the Swarm in-situ data is their latitudinal coverage and the two EIA belts are clearly seen with distribution of std($\Delta N_e$), while HF radars such as JULIA show just their latitude.

b) In 3.3.1, the authors only look at well-developed plumes, while in 3.3.2, all altitudes seem to be taken into account. It is not so clear to me why the authors did not only include high-altitude data in part 3.3.2 if the intention was to directly compare distributions of plumes/SF that can be detected by Swarm? Could the authors comment on this, and eventually clarify their main intention in the text?

**Response:** To clarify, in both sections 3.3.1 and 3.3.2, all altitudes were taken into account during the statistical analysis.

– For section 3.3.1, it was stated in Pg. 11 L.1 - 2 that, "For each RTI plot, the SNR corresponding to the peak height was determined and an event was identified as a significant irregularity when the peak height was 400 km. For peak height less than 400 km, these were classified as weak or no irregularities."

The peak height ≤ 400 km is a representation of bottom-type, bottom-side, and no equatorial spread Fs. This is because Bottom-type layers do not cause strong ionogram spread F or intense radio scintillation at VHF frequencies and above.

Their disturbance in the ionosphere is also not sufficient to cause signatures on air-glows (Hysell, 2000). Also, Bottom-side structures last for only a few hours (stated in Pg. 6 L. 21-25). Therefore, all altitudes were taken into account during the analysis for both sections 3.3.1 and 3.3.2. To make this clear, the following statement has been included in the revised manuscript:

– It is important to note that the peak height less than 400 km is a representation of bottom-type, bottom-side, and no equatorial spread Fs and therefore all spread-F altitudes were taken into consideration during the analysis. (see Pg. 11 L. 2-4)

c) The inclusion of IBI is interesting. However, the authors compare an occurrence rate (IBI) versus std(Ne). I don't understand how this comparison is meaningful, and how the authors can claim "The binned percentage occurrence of IBI = 1 shown in Fig. 12 shows similar results to the ones shown in Fig. 11, but with lower percentage occurrence" (P18 L6). Could the authors please elaborate?

**Response:** As stated in Pg. 14 L. 5-6, the occurrence rate of ionospheric irregularities does not always correspond to the highest amplitude of irregularity structures (Wan et al., 2018). Therefore, it was appropriate to present the distribution of $std(\Delta N_e)$ in Figure 11.

– Note:The climatology maps in Figure 12 have been regenerated in the revised manuscript to clearly show the QLat versus LT trend in percentage occurrence of IBI = 1. Figure 12 in the 1st review of the manuscript was not correct. We were now able to correct the mistake, thanks to the referee raising this point.

To clarify and elaborate, the statements in Pg. 15 L. 11-17 in the revised manuscript have been replaced with the following statements:

– The binned percentage occurrence of IBI = 1 shown in Fig. 12 has similar seasonal characteristics as the in situ irregularities shown in Fig. 11. The percentage occurrence of IBI = 1 ranges from 0 to 20% which is relatively low. The low percentage occurrence of ionospheric irregularities derived from IBI was also observed by Wan et al. (2018). This may be because the magnitude of ionospheric irregularities must be large enough to cause magnetic field fluctuations (Wan et al., 2018). The latitudinal profile of $std(\Delta N_e)$ and percentage occurrence of IBI = 1 have peaks near the anomaly crests (about $\pm 15°$ QLat). The diamagnetic effect in fluctuations of $N_e$ are believed to be the cause of IBI = 1 and this occurs at the anomaly crests (Stolle et al., 2006; Lühr et al., 2003).

**Minor Comments**

0. I found several places where sentences were the same as in the sources e.g.:

P4 L20-21: "is programmed to emit pulses of 26.6 $\mu$s duration at a repetition rate of 100 Hz"

**Response:** It is important to note that the pulse width used in the JULIA experiments during the period of study was 25 $\mu$s and the pulse repetition was 160 pulses per second. The statement identified by the referee refers to the status of JULIA in 2003 and therefore, the statement has been replaced with the following sentence in the revised manuscript to reflect the current operation status of JULIA:

– The pulse width used in the JULIA experiments during the period of study was 25 $\mu$s and the pulse repetition was 160 pulses per second. (see Pg. 4 L. 20-21)

P4 L21-22 "This allows sampling of 200 range gates of 4 km extent from 95 to about 900 km altitude above sea level (Hysell and Burcham, 1998)."

**Response:** For the JULIA radar, 248 range gates of 3.75 km separation were sampled, going from 0 km to about 930 km during the period of study. The statement identified by the referee refers back to the status of JULIA in 1998 and therefore, the statement has been replaced with the following sentence in the revised manuscript to reflect the current operation status of JULIA:

- In addition, 248 range gates of 3.75 km separation were sampled, going from 0 km to about 930 km during the period of study. (see Pg. 4 L. 21-22)

**P6 L5-6** "The spread range at each frequency is the virtual height difference between the top and bottom of the spread echo at a particular frequency

**Response:** The statements in Pg. 5 L. 33-37 of the previous version of the manuscript have been rephrased to:

- The spread-F index QF is defined as the extent of the diffuse reflection in km averaged over all frequencies where a diffuse echo appeared. For simplicity, the virtual height is used at each frequency to determine the range extent of the reflection. The ARTIST software for data analysis is described by Galkin et al. (2008). Spread F ionograms were similarly studied by Abdu et al. (2012) using magnetically conjugate ionosondes in South America, and by Zhang et al. (2015) for ionosondes and scintillation receivers at Sanya. (see Pg. 5 L. 31-36)

**P7 L8-9** (see minor comment 6.)

**Response:** This has been addressed in response to minor comment 6.

This is easily rectified, but fairly important, and I would advise to go through the manuscript once more in detail to check for parts that are too similar to previous articles.

**Response:** We have checked through out the revised manuscript and corrected for texts which are very similar to previous studies.

1. P2 l5 and l11: Precise "equatorial", as there are irregularities at high latitudes also.

   **Response:** This has been changed as suggested. (see Pg. 2 L. 6 and Pg. 2 L. 11 )

2. P2 l15: «(Tsunoda et al. 1982)»: Is this the correct reference? (no mention of the smaller extent, it seems).

   **Response:** The reference has been replaced with: (Lühr et al., 2014; Xiong et al., 2016; Rino et al., 2016) (see Pg. 2 L. 15)

3. P5 l12: "(Ngwira et al., 2013a)": Is this the correct reference?

   **Response:** The correct reference is (Ngwira et al. 2013). This has been adjusted accordingly in the revised manuscript. (see Pg. 5 L. 6 and all through out the revised manuscript)

4. P5 l26: "Field-aligned irregularities in the F-region are often observed between 1800 LT and 0600 LT by the JULIA radar": Add reference.

   **Response:** The references (Hysell and Burcham, 1998; Smith et al., 2016) have been added in the revised manuscript. (see Pg. 5 L. 25)

5. P6 l17: "certainly": I am not sure it is certain. Maybe "most likely", or "probably", or some alternative.

   **Response:** The word "certainly" has been replaced with "most likely" as suggested. (see Pg. 6 L. 9)

6. P7 l8-9: "Bottom-type layers do not cause strong ionogram spread F or intense radio scintillation at VHF frequencies and above" same sentence as in Hysell, 2000 except with one word removed. Modify slightly and add reference.

   **Response:** The statement has been rephrased to:

   - Bottom-type structures are too weak to induce prominent ionogram spread F or cause intense radio scintillation at VHF frequencies and above (Hysell, 2000). (see Pg. 6 L. 18-19)

7. P11 l5-12: A few points are still unclear:

a) how were the conjunctions selected with respect to time?

b) By peak height, is it meant "the altitude with largest SNR?

c) RTI has been defined as Rayleigh - Taylor Instability P.2 L.10

**Response:**

a) We restricted the comparison to times when Swarm was within $\pm 5°$longitude of JULIA. An example is also shown in panel (i) of Fig. 4 for conjunctions between Swarm and JULIA at specific time range. (see Pg 10 L. 11-12)

b)The peak height corresponds to the maximum range in the Range-Time-Intensity plot where SNR was recorded (stated in Pg. 6 L. 27-28). This is not necessarily, the largest SNR.

c) To make this clear, the RTI referring to the JULIA observations has been replaced with "Range-Time-Intensity" in the revised manuscript.(see Pg. 10 L. 17 and all through out the revised manuscript)

8. P12 l14-15: "This suggests that plume structures may not have ascended to Swarm altitudes by the time the satellites passed...": Couldn't this actually be checked precisely, since the authors wrote they made summary plots like figure 4 for each pass?

**Response:** We confirm that we checked on the Swarm altitudes during the pass after making summary plots like those in Fig. 4. Indeed we observed that for cases when JULIA and ionosonde recorded irregularity signatures while, Swarm never encountered any structures, the plume structures existed at altitudes less than that of the Swarm orbits by the time the satellites passed over Jicamarca.

To make this clear, the statement has been rephrased to:

– For these cases, the Swarm altitudes during the pass were examined. It was observed that the plume structures did not ascended to Swarm altitudes by the time the satellites passed over Jicamarca or the satellites were simply in a different location. (see Pg. 11 L.22-24)

9. P15 l5: "geomagnetic conditions": Is it still Kp<3?

**Response:** The geomagnetic conditions considered are when $Kp \leq 3$. To make this clear, this has been indicated in the revised manuscript. (See Pg. 13 L. 8)

10. P19 l9: "Therefore, we presented the calculated std(Ne) per bin" : This is still unclear to me. Do the authors show the average std(Ne) per bin for the different periods, or the maxima, or do they only have one std(Ne) value per bin?

**Response:** A single std(dNe) value was calculated per bin during the analysis.

11. P17 l7: Define IBI.

**Response:** IBI is defined as the Ionospheric Bubble Index. This has been added in the revised manuscript. (see Pg. 15 L. 4)

12. P18 l2: The binning is different with respect to the one made for std(Ne). Is there a reason for this?

**Response:** This was wrongly stated. The same binning was used for both IBI and std(dNe). To correct this the statement has been changed to:

– For each season, the data was then binned into $1° \times 0.1$ hr quasi-dipole latitude-local time bin. (see Pg. 14 L. 4 and Pg. 15 L. 9)

13. P18 l7: add "(. . . )" to Wan et al. 2018

   **Response:** This has been changed as suggested. (see Pg. 15 L.15 )

14. P19 l29-32: "on . . . . Soltice": about three months of data were used to represent the conditions at equinox or solstice, so the authors may consider to write a bit more precisely.

   **Response:** To clarify, the three months used for each season were already defined in Pg. 12 L. 27-28.

[revised manuscript text omitted]